# How Powerful are Performance Predictors in Neural Architecture Search?

**Colin White**[1]*, **Arber Zela**[2], **Binxin Ru**[3], **Yang Liu**[1], **Frank Hutter**[2,4]
[1] Abacus.AI, [2] University of Freiburg, [3] University of Oxford,
[4] Bosch Center for Artificial Intelligence

## Abstract

Early methods in the rapidly developing field of neural architecture search (NAS) required fully training thousands of neural networks. To reduce this extreme computational cost, dozens of techniques have since been proposed to predict the final performance of neural architectures. Despite the success of such performance prediction methods, it is not well-understood how different families of techniques compare to one another, due to the lack of an agreed-upon evaluation metric and optimization for different constraints on the initialization time and query time. In this work, we give the first large-scale study of performance predictors by analyzing 31 techniques ranging from learning curve extrapolation, to weight-sharing, to supervised learning, to zero-cost proxies. We test a number of correlation- and rank-based performance measures in a variety of settings, as well as the ability of each technique to speed up predictor-based NAS frameworks. Our results act as recommendations for the best predictors to use in different settings, and we show that certain families of predictors can be combined to achieve even better predictive power, opening up promising research directions. Our code, featuring a library of 31 performance predictors, is available at `https://github.com/automl/naslib`.

## 1 Introduction

Neural architecture search (NAS) is a popular area of machine learning, which aims to automate the process of developing neural architectures for a given dataset. Since 2017, a wide variety of NAS techniques have been proposed [78, 45, 32, 49]. While the first NAS techniques trained thousands of architectures to completion and then evaluated the performance using the final validation accuracy [78], modern algorithms use more efficient strategies to estimate the performance of partially-trained or even untrained neural networks [11, 2, 54, 34, 38].

Recently, many performance prediction methods have been proposed based on training a model to predict the final validation accuracy of an architecture just from an encoding of the architecture. Popular choices for these models include Gaussian processes [60, 17, 51], neural networks [36, 54, 65, 69], tree-based methods [33, 55], and so on. However, these methods often require hundreds of fully-trained architectures to be used as training data, thus incurring high initialization time. In contrast, learning curve extrapolation methods [11, 2, 20] need little or no initialization time, but each individual prediction requires partially training the architecture, incurring high query time. Very recently, a few techniques have been introduced which are fast both in query time and initialization time [38, 1], computing predictions based on a single minibatch of data. Finally, using shared weights [45, 4, 32] is a popular paradigm for NAS [73, 25], although the effectiveness of these methods in ranking architectures is disputed [53, 74, 76].

Despite the widespread use of performance predictors, it is not known how methods from different families compare to one another. While there have been some analyses on the best predictors within

---

*{colin, yang}@abacus.ai, {zelaa, fh}@cs.uni-freiburg.de, robin@robots.ox.ac.uk

35th Conference on Neural Information Processing Systems (NeurIPS 2021).

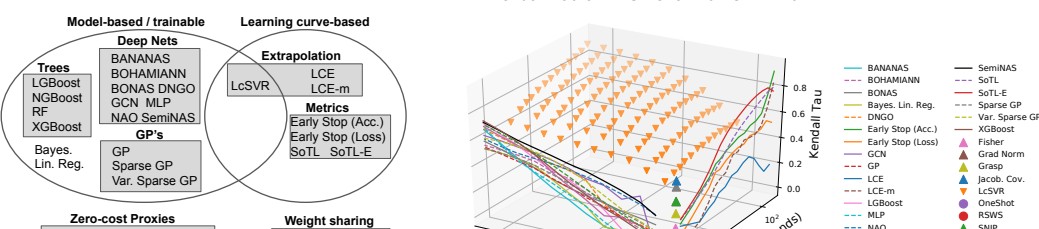

Figure 1: Categories of performance predictors (left). Kendall Tau rank correlation for performance predictors with respect to initialization time and query time (right). Each type of predictor is plotted differently based on whether it allows variable initialization time and/or variable query time. For example, the sixteen model-based predictors have a fixed query time and variable initialization time, so they are plotted as curves parallel to the X-Z plane.

each class [41, 72], for many predictors, the only evaluation is from the original work that proposed the method. Furthermore, no work has previously compared the predictors *across* different families of performance predictors. This leads to two natural questions: how do zero-cost methods, model-based methods, learning curve extrapolation methods, and weight sharing methods compare to one another across different constraints on initialization time and query time? Furthermore, can predictors from different families be combined to achieve even better performance?

In this work, we answer the above questions by giving the first large-scale study of performance predictors for NAS. We study 31 predictors across four popular search spaces and four datasets: NAS-Bench-201 [13] with CIFAR-10, CIFAR-100, and ImageNet16-120, NAS-Bench-101 [71] and DARTS [32] with CIFAR-10, and NAS-Bench-NLP [21] with Penn TreeBank. In order to give a fair comparison among different classes of predictors, we run a full portfolio of experiments, measuring the Pearson correlation and rank correlation metrics (Spearman, Kendall Tau, and sparse Kendall Tau), across a variety of initialization time and query time budgets. We run experiments using a training and test set of architectures generated both uniformly at random, as well as by mutating the highest-performing architectures (the latter potentially more closely resembling distributions encountered during an actual NAS run). Finally, we test the ability of each predictor to speed up NAS algorithms, namely Bayesian optimization [36, 54, 69, 51] and predictor-guided evolution [66, 59].

Since many predictors so far had only been evaluated on one search space, our work shows which predictors have consistent performance across search spaces. Furthermore, by conducting a study with three axes of comparison (see Figure 1), and by comparing various types of predictors, we see a more complete view of the state of performance predictor techniques that leads to interesting insights. Notably, we show that the performance of predictors from different families are complementary and can be combined to achieve significantly higher performance. The success of these experiments opens up promising avenues for future work.

Overall, our experiments bridge multiple areas of NAS research and act as recommendations for the best predictors to use under different runtime constraints. Our code, based on the NASLib library [52], can be used as a testing ground for future performance prediction techniques. In order to ensure reproducibility of the original results, we created a table to clarify which of the 31 predictors had previously published results on a NAS-Bench search space, and how these published results compared to our results (Table 7). We also adhere to the NeurIPS 2021 checklist along with the specialized NAS best practices checklist [31].

**Our contributions.** We summarize our main contributions below.

- We conduct the first large-scale study of performance predictors for neural architecture search by comparing model-based methods, learning curve extrapolation methods, zero-cost methods, and weight sharing methods across a variety of settings.
- We release a comprehensive library of 31 performance predictors on four different search spaces.
- We show that different families of performance predictors can be combined to achieve substantially better predictive power than any single predictor.

## 2   Related Work

NAS has been studied since at least the 1990s [19, 58], and has been revitalized in the last few years [78]. While initial techniques focused on reinforcement learning [78, 45] and evolutionary search [37, 49], one-shot NAS algorithms [32, 12, 4] and predictor-based NAS algorithms [65, 54, 69] have recently become popular. We give a brief survey of performance prediction techniques in Section 3. For a survey on NAS, see [15]. The most widely used type of search space in prior work is the cell-based search space [79], where the architecture search is over a relatively small directed acyclic graph representing an architecture.

A few recent works have compared different performance predictors on popular cell-based search spaces for NAS. Siems et al. [55] studied graph neural networks and tree-based methods, and found that gradient-boosted trees and graph isomorphism networks performed the best. However, the comparison was only on a single search space and dataset, and the explicit goal was to achieve maximum performance given a training set of around 60 000 architectures. Another recent paper [41] studied various aspects of supernetwork training, and separately compared four model-based methods: random forest, MLP, LSTM, and GATES [42]. However, the comparisons were again on a single search space and dataset and did not compare between multiple families of performance predictors. Other papers have proposed new model-based predictors and compared the new predictors to other model-based baselines [34, 65, 54, 69]. Finally, a recent paper analyzed training heuristics to make weight-sharing more effective at ranking architectures [72]. To the best of our knowledge, no prior work has conducted comparisons across multiple families of performance predictors.

## 3   Performance Prediction Methods for NAS

In NAS, given a search space $\mathcal{A}$, the goal is to find $a^* = \mathrm{argmin}_{a \in \mathcal{A}} f(a)$, where $f$ denotes the validation error of architecture $a$ after training on a fixed dataset for a fixed number of epochs $E$. Since evaluating $f(a)$ typically takes hours (as it requires training a neural network from scratch), many NAS algorithms make use of performance predictors to speed up this process. A *performance predictor* $f'$ is defined generally as any function which predicts the final accuracy or ranking of architectures, without fully training the architectures. That is, evaluating $f'$ should take less time than evaluating $f$, and $\{f'(a) \mid a \in \mathcal{A}\}$ should ideally have high correlation or rank correlation with $\{f(a) \mid a \in \mathcal{A}\}$.

Each performance predictor is defined by two main routines: an **initialization** routine which performs general pre-computation, and a **query** routine which performs the final architecture-specific computation: it takes as input an architecture specification, and outputs its predicted accuracy. For example, one of the simplest performance predictors is early stopping: for any **query**$(a)$, train $a$ for $E/2$ epochs instead of $E$ [77]. In this case, there is no general pre-computation, so initialization time is zero. On the other hand, the query time for each input architecture is high because it involves training the architecture for $E/2$ epochs. In fact, the runtime of the initialization and query routines varies substantially based on the type of predictor. In the context of NAS algorithms, the initialization routine is typically performed once at the start of the algorithm, and the query routine is typically performed many times throughout the NAS algorithm. Some performance predictors also make use of an **update** routine, when part of the computation from initialization needs to be updated without running the full procedure again (for example, in a NAS algorithm, a model may be updated periodically based on newly trained architectures). Now we give an overview of the main families of predictors. See Figure 1 (left) for a taxonomy of performance predictors.

**Model-based (trainable) methods.**   The most common type of predictor, the model-based predictor, is based on supervised learning. The initialization routine consists of fully training many architectures (i.e., evaluating $f(a)$ for many architectures $a \in \mathcal{A}$) to build a training set of datapoints $\{a, f(a)\}$. Then a model $f'$ is trained to predict $f(a)$ given $a$. While the initialization time for model-based predictors is very high, the query time typically takes less than a second, which allows thousands of predictions to be made throughout a NAS algorithm. The model is also updated regularly based on the new datapoints. These predictors are typically used within BO frameworks [36, 54], evolutionary frameworks [66], or by themselves [67], to perform NAS. Popular choices for the model include tree-based methods (where the features are the adjacency matrix representation of

the architectures) [33, 55], graph neural networks [36, 54], Gaussian processes [47, 51], and neural networks based on specialized encodings of the architecture [69, 42].

**Learning curve-based methods.**  Another family predicts the final performance of architectures using only a partially trained network, by extrapolating the learning curve. This is accomplished by fitting the partial learning curve to an ensemble of parametric models [11], or by simply summing the training losses observed so far [50]. Early stopping as described earlier is also a learning curve-based method. Learning curve methods do not require any initialization time, yet the query time typically takes minutes or hours, which is orders of magnitude slower than the query time in model-based methods. Learning curve-based methods can be used in conjunction with multi-fidelty algorithms, such as Hyperband or BOHB [27, 16, 24].

**Hybrid methods.**  Some predictors are hybrids between learning curve and model-based methods. These predictors train a model at initialization time to predict $f(a)$ given both $a$ and a partial learning curve of $a$ as features. Models in prior work include an SVR [2], or a Bayesian neural network [20]. Although the query time and initialization time are both high, hybrid predictors tend to have strong performance.

**Zero-cost methods.**  Another class of predictors have no initialization time and very short query times (so-called "zero-cost" methods). These predictors compute statistics from just a single forward/backward propagation pass for a single minibatch of data, by computing the correlation of activations within a network [38], or by adapting saliency metrics proposed in pruning-at-initialization literatures [23, 1]. Similar to learning curve-based methods, since the only computation is specific to each architecture, the initialization time is zero. Zero-cost methods have recently been used to warm start NAS algorithms [1].

**Weight sharing methods.**  Weight sharing [45] is a popular approach to substantially speed up NAS, especially in conjunction with a one-shot algorithm [32, 12]. In this approach, all architectures in the search space are combined to form a single over-parameterized supernetwork. By training the weights of the supernetwork, all architectures in the search space can be evaluated quickly using this set of weights. To this end, the supernetwork can be used as a performance predictor. This results in NAS algorithms [32, 28] which are significantly faster than sequential NAS algorithms, such as evolution or Bayesian optimization. Recent work has shown that although the shared weights are sometimes not effective at ranking architectures [53, 74, 76], one-shot NAS techniques using shared weights still achieve strong performance [73, 25].

**Tradeoff between intialization and query time.**  The main families mentioned above all have different initialization and query times. The tradeoffs between initialization time, query time, and performance depend on a few factors such as the type of NAS algorithm and its total runtime budget, and different settings are needed in different situations. For example, if there are many architectures whose performance we want to estimate, then we should have a low query time, and if we have a high total runtime budget, then we can afford a high initialization time. We may also change our runtime budget throughout the run of a single NAS algorithm. For example, at the start of a NAS algorithm, we may want to have coarse estimates of a large number of architectures (low initialization time, low query time such as zero-cost predictors). As the NAS algorithm progresses, it is more desirable to receive higher-fidelity predictions on a smaller set of architectures (model-based or hybrid predictors). The exact budgets depend on the type of NAS algorithm.

**Choice of performance predictors.**  We analyze 31 performance predictors defined in prior work: BANANAS [69], Bayesian Linear Regression [6], BOHAMIANN [57], BONAS [54], DNGO [56], Early Stopping with Val. Acc. (e.g. [77, 27, 16, 79]) Early Stopping with Val. Loss. [50], Fisher [1], Gaussian Process (GP) [48], GCN [75], Grad Norm [1], Grasp [64], Jacobian Covariance [38], LCE [11], LCE-m [20], LcSVR [2], LGBoost/GBDT [33], MLP [69], NAO [35], NGBoost [55], OneShot [73], Random Forest (RF) [55], Random Search with Weight Sharing (RSWS) [26], SemiNAS [34], SNIP [23], SoTL [50], SoTL-E [50], Sparse GP [3], SynFlow [61], Variational Sparse GP [63], and XGBoost [55]. For any method that did not have an architecture encoding already defined (such as the tree-based methods, GP-based methods, and Bayesian Linear Regression), we use the standard adjacency matrix encoding, which consists of the adjacency matrix of the architecture along with a one-hot list of the operations [71, 68]. By open-sourcing our code, we encourage

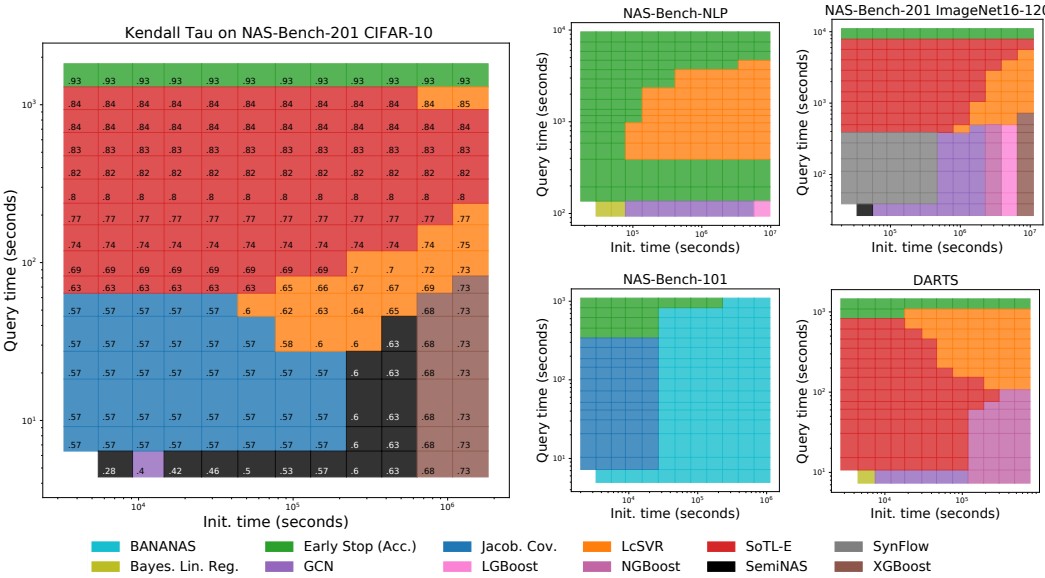

Figure 2: The performance predictors with the highest Kendall Tau values for all initialization time and query time budgets on NAS-Bench-201, NAS-Bench-101, NAS-Bench-NLP and DARTS. For example, on NAS-Bench-201 CIFAR-10 (left) with an initialization time of $10^6$ seconds and query time of 10 seconds, XGBoost achieves a Kendall Tau value of .73 which is the highest value out of the 31 predictors that we tested at that budget.

implementing more (existing and future) performance predictors which can then be compared to the 31 which we focus on in this work. In Section B.1, we give descriptions and detailed implementation details for each performance predictor. In Section D, we give a table that describes for which predictors we were able to reproduce published results, and for which predictors it is not possible (e.g., since some predictors were released before the creation of NAS benchmarks).

## 4 Experiments

We now discuss our experimental setup and results. We discuss reproducibility in Sections A and D, and our code (based on the NASLib library [52]) is available at `https://github.com/automl/naslib`. We split up our experiments into two categories: evaluating the performance of each predictor with respect to various correlation metrics (Section 4.1), and evaluating the ability of each predictor to speed up predictor-based NAS algorithms (Section 4.2). We start by describing the four NAS benchmarks used in our experiments.

**NAS benchmark datasets.** NAS-Bench-101 [71] consists of over $423\,000$ unique neural architectures with precomputed training, validation, and test accuracies after training for 4, 12, 36, and 108 epochs on CIFAR-10 [71]. The cell-based search space consists of five nodes which can take on any directed acyclic graph (DAG) structure, and each node can be one of three operations. Since learning curve information is only available at four epochs, it is not possible to run most learning curve extrapolation methods on NAS-Bench-101. NAS-Bench-201 [13] consists of $15\,625$ architectures (out of which $6\,466$ are unique after removing isomorphisms [13]). Each architecture has full learning curve information for training, validation, and test losses/accuracies for 200 epochs on CIFAR-10 [22], CIFAR-100, and ImageNet-16-120 [10]. The search space consists of a cell which is a complete DAG with 4 nodes. Each edge can take one of five different operations. The DARTS search space [32] is significantly larger with roughly $10^{18}$ architectures. The search space consists of two cells, each with seven nodes. The first two nodes are inputs from previous layers, and the intermediate four nodes can take on any DAG structure such that each node has two incident edges. The last node is the output node. Each edge can take one of eight operations. In our experiments, we make use of the training data from NAS-Bench-301 [55], which consists of $23\,000$ architectures drawn uniformly at random and trained on CIFAR-10 for 100 epochs. Finally, the NAS-Bench-NLP search space [21] is even

larger, at $10^{53}$ LSTM-like cells, each with at most 25 nodes in any DAG structure. Each cell can take one of seven operations. In our experiments, we use the NAS-Bench-NLP dataset, which consists of 14 000 architectures drawn uniformly at random and trained on Penn Tree Bank [40] for 50 epochs.

**Hyperparameter tuning.** Although we used the code directly from the original repositories (sometimes making changes when necessary to adapt to NAS-Bench search spaces), the predictors had significantly different levels of hyperparameter tuning. For example, some of the predictors had undergone heavy hyperparameter tuning on the DARTS search space (used in NAS-Bench-301), while other predictors (particularly those from 2017 or earlier) had never been run on cell-based search spaces. Furthermore, most predictor-based NAS algorithms can utilize cross-validation to tune the predictor periodically throughout the NAS algorithm. This is because the bottleneck for predictor-based NAS algorithms is typically the training of architectures, not fitting the predictor [56, 30, 16]. Therefore, it is fairer and also more informative to compare performance predictors which have had the same level of hyperparameter tuning through cross-validation. For each search space, we run random search on each performance predictor for 5000 iterations, with a maximum total runtime of 15 minutes. The final evaluation uses a separate test set. The hyperparameter value ranges for each predictor can be found in Section B.2.

### 4.1 Performance Predictor Evaluation

We evaluate each predictor based on three axes of comparison: initialization time, query time, and performance. We measured performance with respect to several different metrics: Pearson correlation and three different rank correlation metrics (Spearman, Kendall Tau, and sparse Kendall Tau [72, 55]). The experimental setup is as follows: the predictors are tested with 11 different initialization time budgets and 14 different query time budgets, leading to a total of 154 settings. On NAS-Bench-201 CIFAR-10, the 11 initialization time budgets are spaced logarithmically from 1 second to $1.8 \times 10^7$ seconds on a 1080 Ti GPU (which corresponds to training 1000 random architectures on average) which is consistent with experiments conducted in prior work [65, 69, 34]. For other search spaces, these times are adjusted based on the average time to train 1000 architectures. The 14 query time budgets are spaced logarithmically from 1 second to $1.8 \times 10^4$ seconds (which corresponds to training an architecture for 199 epochs). These times are adjusted for other search spaces based on the training time and different number of epochs. Once the predictor is initialized, we draw a test set of 200 architectures uniformly at random from the search space. For each architecture in the test set, the predictor uses the specified query time budget to make a prediction. We then evaluate the quality of the predictions using the metrics described above. We average the results over 100 trials for each (initialization time, query time) pair.

**Results and discussion.** Figure 1 shows a full three-dimensional plot for NAS-Bench-201 on CIFAR-10 over initialization time, query time, and Kendall Tau rank correlation. Of the 31 predictors we tested, we found that just seven of them are Pareto-optimal with respect to Kendall Tau, initialization time, and query time. That is, only seven algorithms have the highest Kendall Tau value for at least one of the 154 query time/initialization time budgets on NAS-Bench-201 CIFAR-10. This can be seen more clearly in Figure 2 (left), which is a view from above Figure 1: each lattice point displays the predictor with the highest Kendall Tau value for the corresponding budget. In Figure 2 (right), we plot the Pareto-optimal predictors for five different dataset/search space combinations. In Section B.3, we give the full 3D plots and report the variance across trials for each method. In Figure 4 (left), we also plot the Pearson and Spearman correlation coefficients for NAS-Bench-201 CIFAR-10. The trends between these measures are largely the same, although we see that SemiNAS performs better on the rank-based metrics. For the rest of this section, we focus on the popular Kendall Tau metric, giving the full results for the other metrics in Section B.3.

We see similar trends across DARTS and the two NAS-Bench-201 datasets. NAS-Bench-NLP also has fairly similar trends, although early stopping performs comparatively stronger. NAS-Bench-101 is different from the other search spaces both in terms of the topology and the benchmark itself, which we discuss later in this section.

In the low initialization time, low query time region, Jacobian covariance or SynFlow perform well across NAS-Bench-101 and NAS-Bench-201. However, none of the six zero-cost methods perform well on the larger DARTS search space. Weight sharing (which also has low initialization and low query time, as seen in Figure 1), did not yield high Kendall Tau values for these search spaces, either,

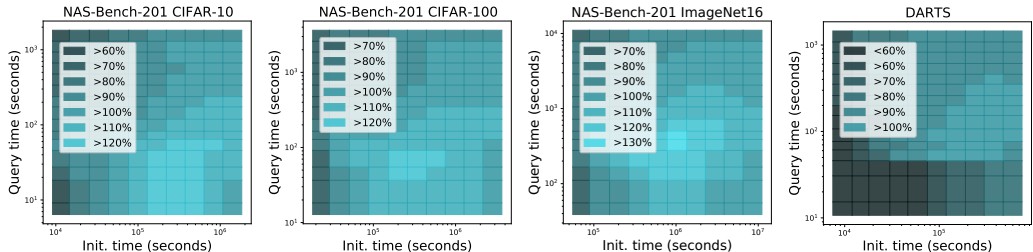

Figure 3: Percentage of OMNI's Kendall Tau value compared to the next-best predictors for each budget constraint.

which is consistent with recent work [53, 74, 76]. However, rank correlation is not as crucial to one-shot NAS algorithms as it is for black box predictor-based methods, as demonstrated by prior one-shot NAS methods that do perform well [25, 73, 32, 28, 12]. In the low initialization time, high query time region, sum of training losses (SoTL-E) consistently performed the best, outperforming other learning-curve based methods.

The high initialization time, low query time region (especially the bottom row of the plots, corresponding to a query time of 1 second) is by far the most competitive region in the recent NAS literature. Sixteen of the 31 predictors had query times under one second, because many NAS algorithms are designed to initialize (and continually update) performance predictors that are used to quickly query thousands of candidate architectures. GCN and SemiNAS, the specialized GCN/semi-supervised methods, perform especially well in the first half of this critical region, when the initialization time is relatively low. However, boosted tree methods actually performed best in the second half of the critical region where the initialization time is high, which is consistent with prior work [33, 55]. Recall that for model-based methods, the initialization time corresponds to training architectures to be used as training data for the performance predictor. Therefore, our results suggest that techniques which can extract better latent features of the architectures can make up for a small training dataset, but methods based purely on performance data work better when there is enough such data.

Perhaps the most interesting finding is that on NAS-Bench-101/201, SynFlow and Jacobian covariance, which take three seconds each to compute, both outperform all model-based methods even after *30 hours* of initialization. Put another way, NAS algorithms that make use of model-based predictors may be able to see substantial improvements by using Jacobian covariance instead of a model-based predictor in the early iterations.

**The Omnipotent Predictor.** One conclusion from Figure 2 is that different types of predictors are specialized for specific initialization time and query time constraints. A natural follow-up question is whether different families are complementary and can be combined to achieve stronger performance. In this section, we run a proof-of-concept to answer this question. We combine the best-performing predictors from three different families in a simple way: the best learning curve method (SoTL-E), and the best zero-cost method (Jacobian covariance), are used as additional input features for a model-based predictor (we separately test SemiNAS and NGBoost). We call this method OMNI, the omnipotent predictor. We give results in Figure 3 and pseudo-code as well as additional experiments in Section B.4. In contrast to all other predictors, the performance of OMNI is strong across almost all budget constraints and search spaces. In some settings, OMNI achieves a Kendall Tau value 30% higher than the next-best predictors.

The success of OMNI verifies that the information learned by different families of predictors are complementary: the information learned by extrapolating a learning curve, by computing a zero-cost proxy, and by encoding the architecture, all improve performance. We further confirm this by running an ablation study for OMNI in Section B.4. We can hypothesize that each predictor type measures distinct quantities: SOTL-E measures the training speed, zero-cost predictors measure the covariance between activations on different datapoints, and model-based predictors simply learn patterns between the architecture encodings and the validation accuracies. Finally, while we showed a proof-of-concept, there are several promising areas for future work such as creating ensembles of the model-based approaches, combining zero-cost methods with model-based methods in more sophisticated ways, and giving a full quantification of the correlation among different families of predictors.

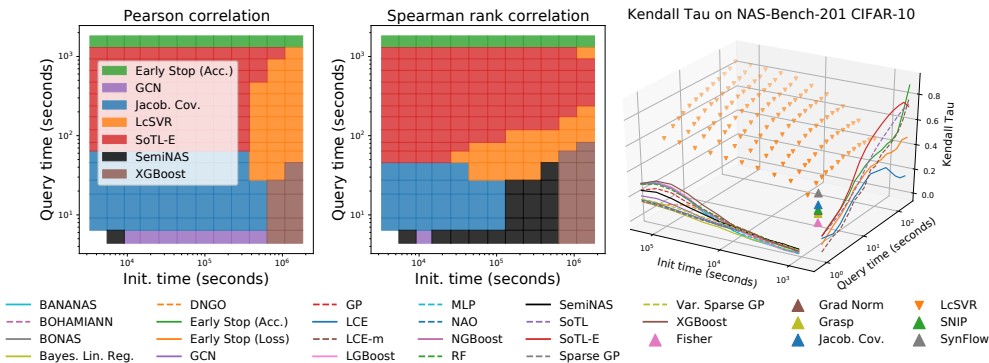

Figure 4: The best predictors on NAS-Bench-201 CIFAR-10 with respect to Pearson (left) and Spearman (middle). Kendall Tau values from a mutation-based training and test set (right).

**NAS-Bench-101: a more complex search space.** In Figure 2, the plot for NAS-Bench-101 looks significantly different than the plots for the other search spaces, for two reasons. The first reason is a technical one: the NAS-Bench-101 API only gives the validation accuracy at four epochs, and does not give the training loss for any epochs. Therefore, we could not implement SoTL or any learning curve extrapolation method. However, all sixteen of the model-based predictors were implemented on NAS-Bench-101. In this case, BANANAS significantly outperformed the next-best predictors (GCN and SemiNAS) across every initialization time. One explanation is due to the complexity of the NAS-Bench-101 search space: while all NAS-Bench-201 architectures have the same graph topology and DARTS architectures' nodes have exactly two incoming edges, the NAS-Bench-101 search space is much more diverse with architectures ranging from a single node and no edges, to five nodes with nine connecting edges. In fact, the architecture encoding used in BANANAS, the path encoding, was designed specifically to deal with the complexity of the NAS-Bench-101 search space (replacing the standard adjacency matrix encoding). To test this explanation, in Appendix B we run several of the simpler tree-based and GP-based predictors using the path encoding, and we see that these methods now surpass BANANAS in performance.

**A mutation-based test set.** The results from Figure 2 used a test set drawn uniformly at random from the search space (and the training set used by model-based predictors was also drawn uniformly at random). However, neighborhood-based NAS algorithms such as local search, regularized evolution, and some versions of Bayesian optimization consider architectures which are local perturbations of the architectures encountered so far. Therefore, the predictors used in these NAS algorithms must be able to distinguish architectures which are local mutations of a small set of seed architectures.

We run an experiment in which the test set is created by mutating architectures from an initial set of seed architectures. Specifically, we draw a set of 50 random architectures and choose the five with the highest validation accuracy as seed architectures. Then we create a set of 200 test architectures by randomly mutating up to three attributes of the seed architectures. Therefore, all architectures in the test set are at most an edit distance of three from a seed architecture, where two architectures are a single edit distance away if they differ by one operation or edge.

We create the training set by randomly choosing architectures from the test set and mutating one random attribute. As in all of our experiments, we ensure that the training set and test set are disjoint. In Figure 4 (right), we plot the correlation results for NAS-Bench-201 CIFAR-10. While the zero-cost and learning curve-based approaches have similar performance, the model-based approaches have significantly worse performance compared to the uniform random setting. This is because the average edit distance between architectures in the test set is low, making it significantly harder for model-based predictors to distinguish the performance of these architectures, even when using a training set that is based on mutations of the test set. In fact, interestingly, the performance of many model-based approaches starts to perform worse after $10^6$ seconds. SemiNAS in particular performs much worse in this setting, and boosted trees have comparatively stronger performance in this setting.

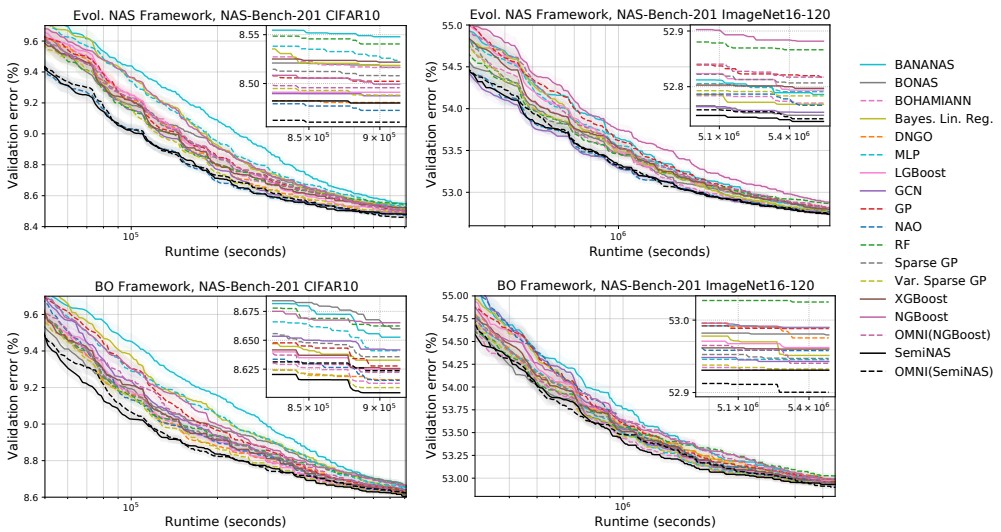

Figure 5: Validation error vs. runtime for the predictor-guided evolution framework and the Bayesian optimization + predictor framework using different predictors.

## 4.2 Predictor-Based NAS Experiments

Now we evaluate the ability of each model-based performance predictor to speed up NAS. We use two popular predictor-based NAS methods: the predictor-guided evolution framework [66, 59], and the Bayesian optimization + predictor framework [17, 36, 54]. The predictor-guided evolution framework is an iterative procedure in which the best architectures in the current population are mutated to create a set of candidate architectures. A predictor (trained on the entire population) chooses $k$ architectures which are then evaluated. In our experiments, the candidate pool is created by mutating the top five architectures 40 times each, and we set $k = 20$. For each predictor, we run predictor-guided evolution for 25 iterations and average the results over 100 trials. The BO + predictor framework is similar to the evolution framework, but an ensemble of three performance predictors are used so that uncertainty estimates for each prediction can be computed. In each iteration, the candidate architectures whose predictions maximize an acquisition function are then evaluated. Similar to prior work, we use independent Thompson sampling [69], as the acquisition function, and an ensemble is created by using a different ordering of the training set and different random weight initializations (if applicable) of the same predictor. In each iteration, the top 20 architectures are chosen from a randomly sampled pool of 200 architectures.

In Figure 5, we present results for both NAS frameworks on NAS-Bench-201 CIFAR-10 and ImageNet16-120 for the 16 model-based predictors. We also test OMNI, using its lowest query time setting (consisting of NGBoost + Jacobian covariance), and another version of OMNI that replaces NGBoost with SemiNAS. Our results show that the model-based predictors with the top Kendall Tau rank correlations in the low query time region from Figure 2 also roughly achieve the best performance when applied for NAS: SemiNAS and NAO perform the best for shorter runtime, and boosted trees perform best for longer runtime. OMNI(NGBoost) consistently outperforms NGBoost, and OMNI(SemiNAS) often achieves top performance. This suggests that using zero-cost methods in conjunction with model-based methods is a promising direction for future study.

## 4.3 So, how powerful are performance predictors?

Throughout Section 4, we tested performance predictors in a variety of settings, by varying the search spaces, datasets, runtime budgets, and training/test distributions. We saw largely the same trends among all of our experiments. Interesting findings included the success of zero-cost predictors even when compared to model-based predictors and learning curve extrapolation predictors with longer runtime budgets, and the fact that information from different families of predictors are complementary. When choosing a performance predictor for new applications, we recommend deciding on a target initialization time and query time budget, consulting Figures 2 and 6, and then combining the best

predictors from the desired runtime setting, similar to OMNI. For example, if a performance predictor with medium initialization time and low runtime is desired for a search space similar to NAS-Bench-201 or DARTS, we recommend using NGBoost with Jacobian covariance and SynFlow as additional features.

## 5    Societal Impact

Our hope is that our work will have a positive impact on the AutoML community by making it quicker and easier to develop and fairly compare performance predictors. For example, AutoML practitioners can consult our experiments to more easily decide on the performance prediction methods best suited to their application, rather than conducting computationally intensive experiments of their own [43]. Furthermore, AutoML researchers can use our library to develop new performance prediction techniques and compare new methods to 31 other algorithms across four search spaces. Since the topic of this work is AutoML, it is a level of abstraction away from real applications. This work may be used to improve deep learning applications, both beneficial (e.g. reducing $CO_2$ emissions), or harmful (e.g. creating language models with heavy bias) to society.

## 6    Conclusions and Limitations

In this work, we gave the first large-scale study of performance predictors for neural architecture search. We compared 31 different performance predictors, including learning curve extrapolation methods, weight sharing methods, zero-cost methods, and model-based methods. We tested the performance of the predictors in a variety of settings and with respect to different metrics. Although we ran experiments on four different search spaces, it will be interesting to extend our experiments to even more machine learning tasks beyond image classification and language modeling.

Our new predictor, OMNI, is the first predictor to combine complementary information from three families of performance preditors, leading to substantially improved performance. While the simplicity of OMNI is appealing, it also opens up new directions for future work by combining different predictors in more sophisticated ways. To facilitate follow-up work, we release our code featuring a library of performance predictors. Our goal is for our repository to grow over time as it is used by the community, so that experiments in our library can be even more comprehensive.

## Acknowledgments and Disclosure of Funding

This work was done while CW and YL were employed at Abacus.AI. AZ and FH acknowledge support by the European Research Council (ERC) under the European Union Horizon 2020 research and innovation programme through grant no. 716721, and by BMBF grant DeToL. BR was supported by the Clarendon Fund of University of Oxford.

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
