# A   NAS Research checklist

There have been a few recent works which have called for improving the reproducibility and fairness in experimental comparisons in NAS research [26, 71, 70]. This led to the release of a NAS best practices checklist [31]. We address each part of the checklist.

1. **Best Practices for Releasing Code**

    For all experiments you report:

    (a) Did you release code for the training pipeline used to evaluate the final architectures? [Yes] We used publicly available NAS benchmarks: NAS-Bench-101, NAS-Bench-201, DARTS/NAS-Bench-301, and NAS-Bench-NLP, so we did not train the architectures ourselves.

    (b) Did you release code for the search space [Yes] Since we used NAS benchmarks, this code is already publicly available.

    (c) Did you release the hyperparameters used for the final evaluation pipeline, as well as random seeds? [Yes] Since we used NAS benchmarks, the final training pipeline was fixed. We released our code, which includes the seeds used.

    (d) Did you release code for your NAS method? [Yes] All of our code is available at `https://github.com/automl/naslib`.

    (e) Did you release hyperparameters for your NAS method, as well as random seeds? [Yes] The hyperparameters are given in Appendix Table 2. We ran 100 trials (random seeds) for each experiment, and we released the code to launch these experiments is in our repository.

2. **Best practices for comparing NAS methods**

    (a) For all NAS methods you compare, did you use exactly the same NAS benchmark, including the same dataset (with the same training-test split), search space and code for training the architectures and hyperparameters for that code? [Yes] We only used NAS benchmarks, which means the training details are fixed.

    (b) Did you control for confounding factors (different hardware, versions of DL libraries, different runtimes for the different methods)? [Yes] We only used NAS Benchmarks, which keep these details fixed.

    (c) Did you run ablation studies? [Yes] Our abalation study for OMNI is in Appendix B.4.

    (d) Did you use the same evaluation protocol for the methods being compared? [Yes] We only used NAS Benchmarks, which keep this fixed.

    (e) Did you compare performance over time? [Yes] Our experiments compare performance over initialization time and query time.

    (f) Did you compare to random search? [No] We did not compare to random search, although we used other baselines for performance predictors such as random forests and simple MLPs.

    (g) Did you perform multiple runs of your experiments and report seeds? [Yes] We ran 100 trials of each experiment, and we released the code to launch these experiments is in our repository.

    (h) Did you use tabular or surrogate benchmarks for in-depth evaluations? [Yes] We only used tabular and surrogate benchmarks.

3. **Best practices for reporting important details**

    (a) Did you report how you tuned hyperparameters, and what time and resources this required? [Yes] We reported this information in Section 4.

    (b) Did you report the time for the entire end-to-end NAS method (rather than, e.g., only for the search phase)? [Yes] We present results along three axes: performance, initialization time, and query time.

    (c) Did you report all the details of your experimental setup? [Yes] We did include all details of the setup in Section 4 and Appendix B.

# B   Details from Section 4 (Experiments)

In this section, we give more details from Section 4, and we present more experiments. In Section B.1, we give a short description and implementation details for all 31 predictors. We also mention the licenses for the datasets we use. Next, in Section B.2, we give the details for hyperparameter tuning. Then in Section B.3, we give detailed experimental results for all search spaces. After that, in Section B.4, we give pseudo-code and an ablation study for OMNI.

## B.1   Descriptions and Implementation Details

We describe all 31 methods that we used.

- **BANANAS.** The BANANAS [69] predictor is a model-based predictor consisting of an ensemble of three MLPs. Architectures are encoded using the *path encoding*, which encodes each possible path from the input to the output of the cell as a bit. We use the code from the original repository, but with PyTorch for the MLPs instead of Tensorflow.

- **Bayesian Linear Regression** Bayesian Linear Regression [6] is one of the simplest Bayesian methods, which assumes that the samples are normally distributed, and assumes that the output labels are a linear function of the input features (the architecture encoding in our case). Therefore, the predictions are sampled from a distribution computed in closed form given the observations, the posterior of the linear model parameters and the observation noise. We use the implementation from the pybnn [2] package and the one-hot adjacency matrix encoding.

- **BOHAMIANN.** BOHAMIANN [57] utilizes Bayesian inference via stochastic gradient Hamiltonian Monte Carlo (SGHMC) in order to sample from the estimated posterior of a Bayesian Neural Network. We use the original implementation from the pybnn package and the one-hot adjacency matrix encoding.

- **BONAS.** Bayesian optimization for NAS [54] is a NAS algorithm which makes use of a graph convolutional network (GCN) as a model-based predictor within an outer loop of Bayesian optimization. In this work, we refer to "BONAS" as the GCN predictor. The implementation from the original paper works on a constrained version of the DARTS search space where the normal cell and the reduction cell have the same architecture. We adapted BONAS for the three search spaces using the original encoding style. Specifically, given that the DARTS search space includes both normal and reduction cells, we encoded both in one adjacency matrix by arranging the two cells' adjacency matrices diagonally and zero-padding the rest.

- **DNGO.** Deep Networks for Global Optimization (DNGO) is an implementation of Bayesian optimization using adaptive basis regression using neural networks instead of Gaussian processes to avoid the cubic scaling [56]. We use the adaptive basis regressor as a model-based predictor, using the original code from the pybnn package and the one-hot adjacency matrix encoding.

- **Early Stopping with Val. Acc.** Early Stopping using validation accuracy has been considered in NAS many times (e.g. [77, 27, 16, 79]). It uses the validation accuracy from the most recent epoch trained so far as a proxy for architecture performance.

- **Early Stopping with Val. Loss** Early stopping with validation loss uses the validation loss instead of the validation accuracy [50].

- **Fisher.** Fisher [1] is a zero-cost predictor which computes the sum over all gradients of the activations in a neural network. Fisher builds off of prior work on channel pruning at initialization [62]. We used the implementation from [1].

- **GCN.** The GCN approach for NAS has also been studied by [67]. Although the code was never released, we used an unofficial implementation online from [75]. The encoding strategy follows that of BONAS.

- **GP.** Gaussian Process (GP) [48] is a simple model where every finite number of random variables have a joint Gaussian distribution. It is commonly used as the default model choice

---

[2] https://github.com/automl/pybnn

in Bayesian optimization, and it is fully specified by its mean and covariance functions. The runtime of GPs is cubic in the number of datapoints, because the covariance matrix must be inverted to compute the predictive distribution is computed. We use the `Pyro` implementation [5] and the one-hot adjacency matrix encoding.

- **Grad Norm.** Grad Norm [1] is a zero-cost predictor which sums the Euclidean norm of the gradients of one minibatch of training data. It was used by [1] as a baseline when comparing other zero-cost methods. We used the implementation from [1].

- **Grasp.** Grasp [64] was introduced as a technique to prune network weights based on a saliency metric at initialisation. It improved over SNIP [23] by approximating the change in gradient norm. Later, it was used as a zero-cost predictor in NAS by [1]. We used the implementation from [1].

- **Jacobian covariance.** Jacobian covariance [38] is a zero-cost method which measures the modelling flexibility of a network based on the covariance of its prediction Jacobians with respect to different image inputs. [38] claims that architectures with more flexible prediction at initialization tend to have better test performance after training. We use the original code.

- **LCE.** Learning curve extrapolation (LCE) [11] takes in a partial learning curve as input, and then extrapolates the curve to a chosen epoch. It works by fitting the curve to several parametric models, and choosing the best model using MCMC. We use the original code, but we used a subset of the original parametric models which we found to improve performance.

- **LCE-m.** LCE-m, introduced as a baseline by [20] is an extrapolation method similar to LCE but with a modified loss function that drops LCE's original mechanism of biasing the search to never underestimate the accuracy at the asymptote of the curve. The list of parametric models was also partially changed. Note that this is not to be confused with the learning curve *prediction* method from [20]. We used the original code, but we used a subset of the parametric models which we found to improve performance.

- **LcSVR.** This is a hybrid predictor, which extrapolates the learning curves using a trained $\nu$-SVR (LcSVR) [2]. Specifically, the $\nu$-SVR model takes in a partial learning curve as well as their first and second derivatives, and the training hyperparameters as the inputs. Like other model-based predictors, the $\nu$-SVR model must be trained by using fully evaluated architectures as training data, and like other learning-curve-based predictors, each new query requires partially training the architecture. We use the original code.

- **LGBoost.** Light Gradient Boosting Machine (LightGBM or LGBoost) [18] was designed to be a more lightweight gradient boosting implementation. LGBoost has been used as a model-based predictor both in NAS algorithms [33] and in the creation of NAS-Bench-301 [55]. We followed the implementation of [33] and used the one-hot adjacency matrix encoding. We reduced the minimum number of data points in a leaf to 5 so that LGBoost could run with smaller training set sizes.

- **MLP.** The multilayer perceptron (MLP) predictor is a model-based predictor consisting of a fully-connected neural network, which has been used by prior work as a baseline for comparisons [69, 65]. We use the implementation in BANANAS, in which the default network has width 10 and depth 20. To encode the architecture, we used a one-hot encoding of the adjacecy matrix and list of operations, as in most prior work in NAS [71].

- **NAO.** Neural Architecture Optimization makes use of an encoder-decoder [35]. The original neural architecture is mapped to a continuous representation via an LSTM encoder network. Performance prediction is powered by a feedfoward neural network, and the decoder is built with an LSTM and attention mechanisms. We used the implementation from SemiNAS [34].

- **NGBoost.** Natural gradient boosting (NGBoost) [14] is another gradient-boosted method that uses natural gradients in order to enable uncertainty estimates of the predictions. It has been studied as a model-based predictor in the creation of NAS-Bench-301 [55]. We used the original code and the one-hot adjacency matrix encoding.

- **OneShot.** We derive the OneShot predictor following a similar procedure as in prior work [73]. We first train the weight-sharing model (supernetwork) with normal SGD training. The number of epochs and number of training examples are picked based on a grid search optimizing the Spearman rank correlation on a validation set. After the weight-sharing network is trained, we use it as a predictor for the performance of an architecture by

computing the performance on the validation set using only the subpath in the supernetwork corresponding to that architecture. The rest of the supernetwork is zeroed out.

- **Random Forest.** Random forests [29] consist of ensembles of decision trees. Random forests have been studied as model-based predictors in the creation of NAS-Bench-301 [55]. We use the Scikit-learn implementation [44] and the one-hot adjacency matrix encoding.

- **RSWS.** Random Search with Weight Sharing (RSWS) [26] is used exactly the same as the OneShot predictor when predicting the performance of an architecture. The only difference is the way RSWS optimizes the weight-sharing model. Instead of training it as a single network, at each mini-batch iteration, RSWS uniformly samples one architecture from the search space and updates the weights of *only* that operations corresponding to that architecture in the supernetwork.

- **SemiNAS.** Semi-supervised NAS (SemiNAS) uses semi-supervised learning with the NAO architecture [34]. Specifically, additional synthetic training data is generated and used to train the architecture. The performance of synthetic training data is predicted by the NAO predictor. We use the original implementation, reducing the ratio of real vs. synthetic data to 1:1 and decreasing the number of epochs to 100 to decrease its extreme training time. SemiNAS was originally implemented for NAS-Bench-101 and ProxylessNAS [7] search spaces. We adapted it for NAS-Bench-201 and DARTS using the original encoding style. Encoder/decoder lengths and vocabulary sizes of the NAO predictor were changed accordingly to fit the new search spaces.

- **SNIP.** Single-shot network pruning (SNIP) was first proposed in [23] as a technique to prune network weights at initialisation. SNIP was later adapted by [1] to become a zero-cost predictor for ranking architecture performance. We used the implementation from [39].

- **SoTL.** Sum of training losses (SoTL) [50] is a learning curve-based predictor, which estimates the generalization performance of architectures based on the sum of their training losses over the epochs trained so far. SoTL does not attempt to predict the final validation accuracy as in model-based predictors, but does output a score with high rank correlation with respect to the final validation accuracy. We use the original implementation of SoTL [50].

- **SoTL-E.** Sum of training losses at last epoch E (SoTL-E) [50] is very similar to SoTL, but it only considers the sum over the training mini-batches in the most recent epoch trained so far.

- **Sparse GP.** Compared to classical GPs, Sparse Gaussian Processes (Sparse GPs) [46] scale better to large amounts of data by introducing the so-called inducing variables to summarize the training data. To bypass the expensive marginal likelihood estimation, variational inference is used in order to approximate the posterior distribution. We use the `Pyro` implementation [5] and the one-hot adjacency matrix encoding.

- **SynFlow.** Synaptic Flow (SynFlow) [61] was introduced as a technique to prune network weights at initialisation based on a saliency metric. It improved over SNIP [23] and Grasp [64] by avoiding layer collapse when performing parameter pruning by taking a product of all parameters in the network. It was used as a zero-cost predictor in NAS by [1]. We used the implementation from [1].

- **Variational Sparse GP.** Variational Sparse Gaussian Process (Var. Sparse GP) [63] is similar to the Sparse GP, but it can handle non-Gaussian likelihoods. We use the `Pyro` implementation [5] and the one-hot adjacency matrix encoding.

- **XGBoost.** eXtreme Gradient Boosting (XGBoost) [9] is the first of three gradient-boosted decision tree implementations that we used. XGBoost has been used as a model-based predictor in the creation of NAS-Bench-301 [55]. We used the original code and the one-hot adjacency matrix encoding.

In Table 1, we discuss the licenses for the NAS datasets we used.

## B.2 Hyperparameter Tuning

Now we give the details of hyperparameter tuning. Recall from Section 4 that we run cross-validation on all model-based predictors that we studied, for two reasons. First, we compared 16 model-based predictors directly from the original repositories when possible, but the published hyperparameters

Table 1: Licenses for the datasets that we use.

| Dataset | License | URL |
|---------|---------|-----|
| NAS-Bench-101 | Apache 2.0 | `https://github.com/google-research/nasbench` |
| NAS-Bench-201 | MIT | `https://github.com/D-X-Y/NAS-Bench-201` |
| NAS-Bench-301 | Apache 2.0 | `https://github.com/automl/nasbench301` |
| NAS-Bench-NLP | None | `https://github.com/fmsnew/nas-bench-nlp-release` |

from different methods have significantly different levels of hyperparameter tuning. Therefore, running more hyperparameter tuning for all predictors will help to level the playing field. Second, most predictor-based NAS algorithms can naturally utilize cross-validation during the search to improve performance. This is because the bottleneck for predictor-based NAS algorithms is typically the training of architectures, not fitting the predictor [56, 30, 16]. Therefore, adding cross-validation to model-based predictors is a more realistic setting. Note that the same is not true for other families of performance predictors. For example, LCE methods are typically used to replace fully training architectures during NAS, therefore we would not know the final validation accuracy of these architectures and would not be able to run cross-validation.

Overall, our goal was to run lightweight cross-validation to level the playing field. For each model-based predictor, we chose 3-5 hyperparameters and chose ranges based on their default values from their original repositories. See Table 2. Note that some methods such as the three GP-based methods and the three methods from pybnn (Bayes. Lin. Reg., BOHAMIANN, DNGO) already had cross-validation built in, so we excluded these. In all of our experiments, we ran random search on each performance predictor for 5000 iterations, with a maximum total runtime of 15 minutes.

## B.3 Additional Experiments

In Figure 2, we plotted the predictors which are Pareto-optimal for each initialization and query time budget. In this section, we present the complete results, including 3D plots for all search spaces, as well as separate plots for initialization time and query time vs. Kendall Tau which include the standard deviation of the results from 100 trials. See Figure 6. We put in a reasonable effort to implement all 31 predictors for all search spaces, however, a few of these combinations are omitted. For example, running learning curve methods on NAS-Bench-101 would require training the architectures from scratch, since NAS-Bench-101 does not include full learning curve information.

In general, in Figure 6 we see that the relative performance of the predictors are largely the same across the three datasets from NAS-Bench-201. However, there are clear differences across NAS-Bench-201 CIFAR-10, NAS-Bench-101, and DARTS, even though all three use CIFAR-10 as the dataset. For example, for high initialization time, BANANAS exhibits the highest rank correlation on NAS-Bench-101 but the worst rank correlation on DARTS. As explained in Section 4, this is largely due to the strength of the path encoding specifically on NAS-Bench-101. However, the path encoding does not scale as well on larger search spaces such as DARTS.

In Figure 7, we plot the predictors which are Pareto-optimal for each initialization and query time budget, for three metrics: Pearson correlation, Spearman rank correlation, and sparse Kendall Tau. We also include Kendall Tau for completeness (which we had already presented in Figure 2).

Since Figures 6 and 7 take up one page each already, we plot the results for the final search space, NAS-Bench-NLP, in Figure 8. We see largely the same trends on this repository as well, with a few differences. Since the NAS-Bench-NLP code is written in a much earlier version of PyTorch (1.1), we were unable to implement the zero-cost predictors in NAS-Bench-NLP. Furthermore, the SOTL and SOTL-E predictors do not perform as well as Early Stop (Acc.) for NAS-Bench-NLP. We see that LcSVR performs particularly well on NAS-Bench-NLP (as was also the case with DARTS). This may be because the large size of the search space gives a bigger benefit for hybrid methods which include LCE components, over model-based methods alone.

Finally, recall that in Section 4, we discussed the differences between NAS-Bench-101 and the other search spaces. We mentioned one technical reason (the NAS-Bench-101 API only gives validation accuracies at four epochs, and does not give the training loss for any epochs) and one reason based on the differences in the search space itself: the NAS-Bench-101 search space is more complex than

Table 2: Hyperparameters of the model-based methods and their default values from their original repositories.

| Model | Hyperparameter | Range | Log-transform | Default Value |
|---|---|---|---|---|
| BANANAS | Num. layers | [5, 25] | false | 20 |
| | Layer width | [5, 25] | false | 20 |
| | Learning rate | [0.0001, 0.1] | true | 0.001 |
| BONAS | Num. layers | [16, 128] | true | 64 |
| | Batch size | [32, 256] | true | 128 |
| | Learning rate | [0.00001, 0.1] | true | 0.0001 |
| GCN | Num. layers | [64, 200] | true | 144 |
| | Batch size | [5, 32] | true | 7 |
| | Learning rate | [0.00001, 0.1] | true | 0.0001 |
| | Weight decay | [0.00001, 0.1] | true | 0.0003 |
| LCSVR | Penalty param. | [0.00001, 10] | true | - |
| | Kernel coefficient | [0.00001, 10] | false | - |
| | Frac. support vectors | [0, 1] | false | - |
| LGBoost | Num. leaves | [10, 100] | false | 31 |
| | Learning rate | [0.001, 0.1] | true | 0.05 |
| | Feature fraction | [0.1, 1] | false | 0.9 |
| MLP | Num. layers | [5, 25] | false | 20 |
| | Layer width | [5, 25] | false | 20 |
| | Learning rate | [0.0001, 0.1] | true | 0.001 |
| NAO | Num. layers | [16, 128] | true | 64 |
| | Batch size | [32, 256] | true | 100 |
| | Learning rate | [0.00001, 0.1] | true | 0.001 |
| NGBoost | Num. estimators | [128, 512] | true | 505 |
| | Learning rate | [0.001, 0.1] | true | 0.081 |
| | Max depth | [1, 25] | false | 6 |
| | Max features | [0.1, 1] | false | 0.79 |
| RF | Num. estimators | [16, 128] | true | 116 |
| | Max features | [0.1, 0.9] | true | 0.17 |
| | Min samples (leaf) | [1, 20] | false | 2 |
| | Min samples (split) | [2, 20] | true | 2 |
| SemiNAS | Num. layers | [16, 128] | true | 64 |
| | Batch size | [32, 256] | true | 100 |
| | Learning rate | [0.00001, 0.1] | true | 0.001 |
| XGBoost | Max depth | [1, 15] | false | 6 |
| | Min child weight | [1, 10] | false | 1 |
| | Col sample (tree) | [0, 1] | false | 1 |
| | Learning rate | [0.001, 0.5] | true | 0.3 |
| | Col sample (level) | [0, 1] | false | 1 |

NAS-Bench-201 and DARTS because it allows any graph topology. Therefore, the path encoding is particularly well-suited for NAS-Bench-101. To test this explanation, in Figure 9 we run several of the simpler tree-based and GP-based predictors using the path encoding, and we see that these methods now surpass BANANAS.

## B.4  OMNI Details and Ablation

In this section, we present more details and experiments for OMNI. Recall that OMNI combines strong predictors from three different families: SoTL-E, Jacobian covariance, and either NGBoost or SemiNAS, from the families of learning curve methods, zero-cost methods, and model-based methods, respectively. See Algorithm 1 for pseudo-code. Recall from Section 3 that performance

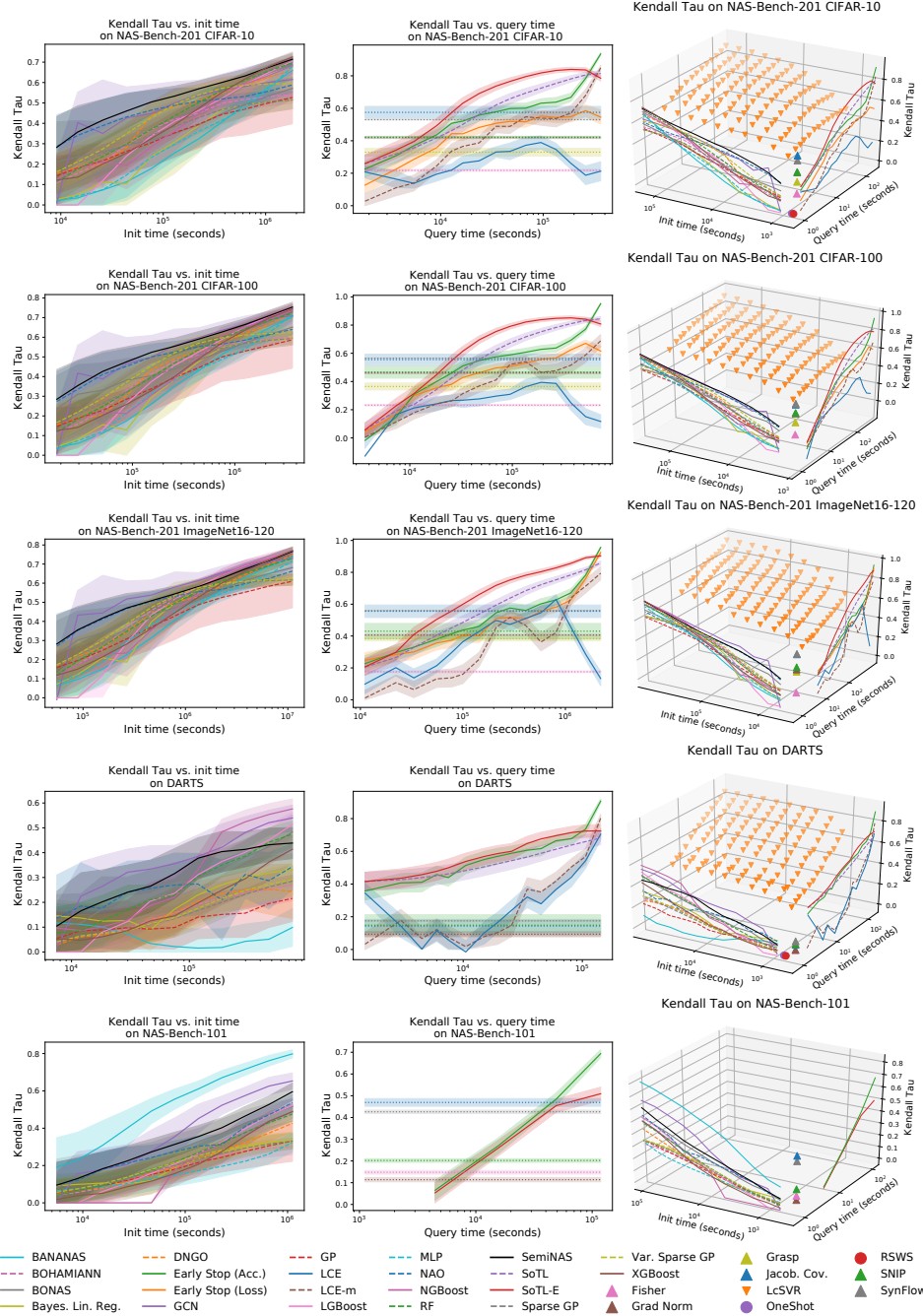

Figure 6: Full results for Kendall Tau with standard deviations shaded.

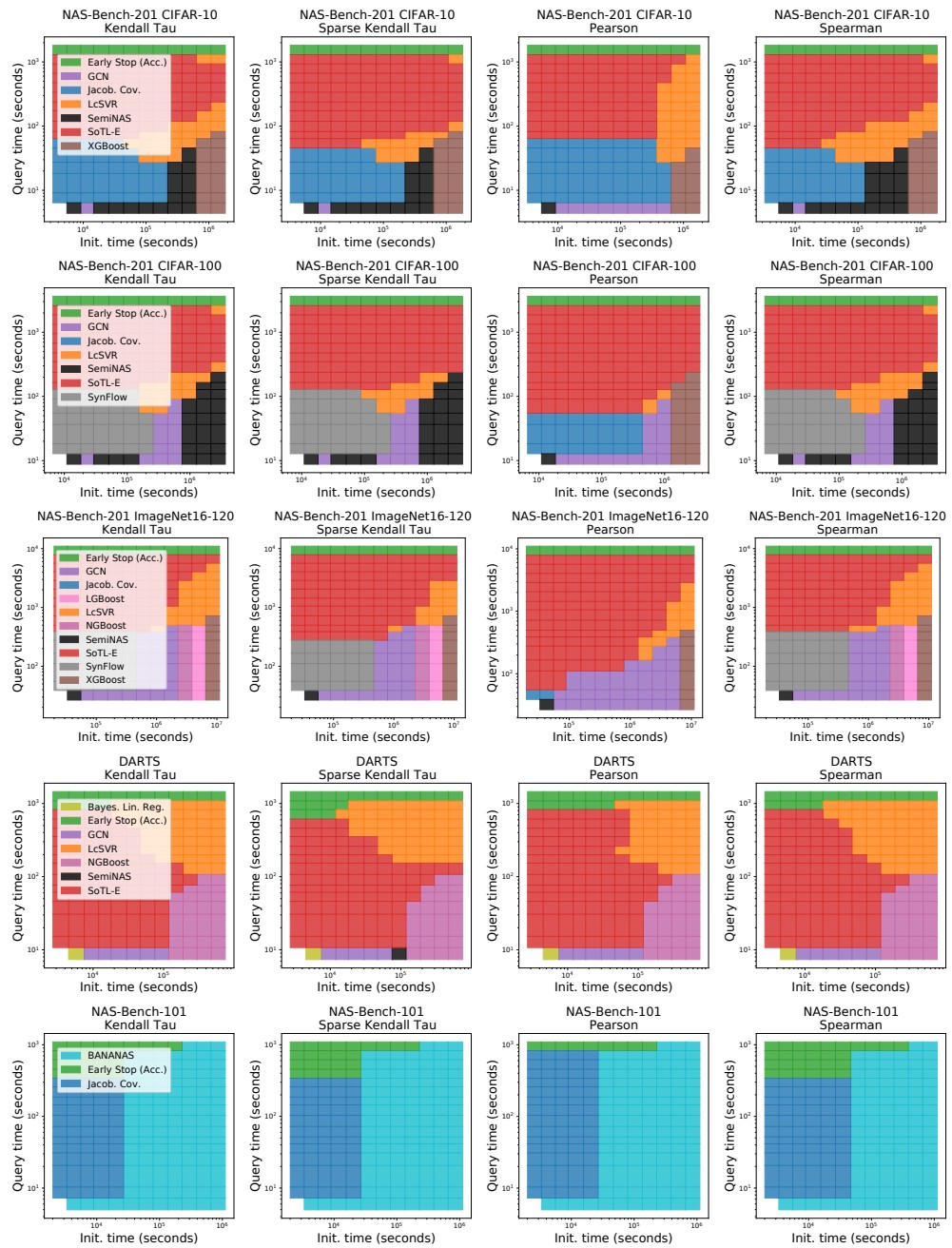

Figure 7: The performance predictors with the highest metrics for all initialization time and query time budgets and search spaces. The first column is repeated from Figure 2.

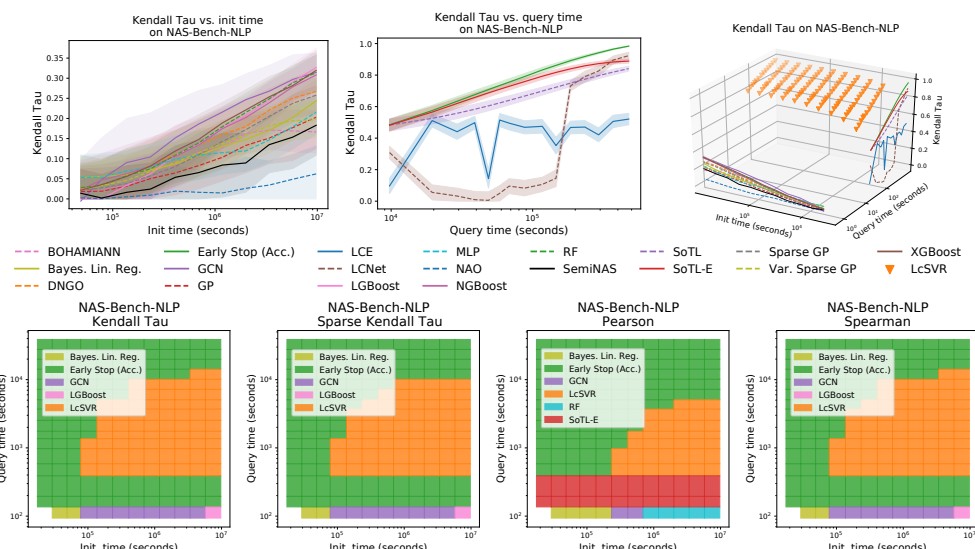

Figure 8: Full results for NAS-Bench-NLP with standard deviations shaded (top row). The performance predictors with the highest metrics for all initialization time and query time budgets on NAS-Bench-NLP (bottom row).

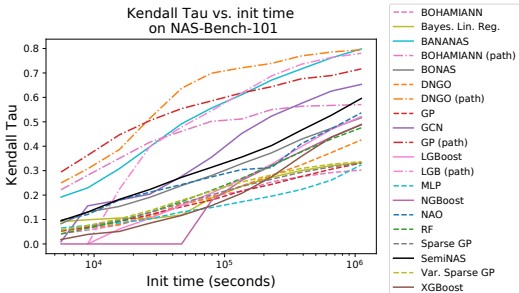

Figure 9: Kendall Tau rank correlation for predictors on NAS-Bench-101. The GP-based and tree-based methods are not competitive until used with the path encoding.

predictors have an initialization stage that is called once, and a query stage that is called many times throughout the NAS algorithm.

Now we give an ablation study for OMNI. We consider four different versions of OMNI: (NGBoost + Jacob. Cov.), (SoTL-E + NGBoost), (Jacob. Cov. + SoTL-E), and (NGBoost + Jacob. Cov. + SoTL-E). Note that Jacob. Cov. + SoTL-E is created by using Jacob. Cov. and SoTL-E as features in NGBoost without the architecture encoding as features. All other OMNI variants use NGBoost with the architecture encodings as features.

In Figure 10, we plot the percentage of Kendall Tau compared to the best predictor in {Jacob. Cov. SoTL-E, and NGBoost} for each initialization time and query time budget, for each OMNI variant. We see that the variants (NGBoost + Jacob. Cov.) and (SoTL-E + NGBoost) both give 20% Kendall Tau improvements for some budget constraints, but do not perform well for other budget constraints. On the other hand, (Jacob. Cov. + SoTL-E) has fairly consistent performance across all budget constraints. Finally, (NGBoost + Jacob. Cov. + SoTL-E) has consistent performance everywhere and peaks at 30% improvement. This ablation study shows that predictors from all three families are needed to achieve maximum performance.

Next, we give additional experiments for OMNI in other settings. Note that Figure 3 used the version of OMNI with NGBoost. See Figure 11 for the version of OMNI with SemiNAS, which performs worse. Finally, see Figure 12 for the performance of OMNI in the mutation-based setting described in Section 4. Note that it performs comparatively worse than in the standard uniformly random setting.

**Algorithm 1** OMNI predictor

---

**Input:** Search space $A$, dataset $D$, initialization time budget $B_{\text{init}}$, query time budget $B_{\text{query}}$.

**Initialization():**

- $\mathcal{D}_{\text{train}} \leftarrow \emptyset$
- While $t < B_{\text{init}}$
  - Draw an architecture $a$ randomly from $A$
  - Train $a$ to completion to compute val. accuracy $y_a$
  - $\mathcal{D}_{\text{train}} \leftarrow \mathcal{D}_{\text{train}} \cup \{(a, y_a)\}$
- Train an NGBoost model $m$ to predict the final val. accuracy of architectures from $\mathcal{D}_{\text{train}}$, using the architecture encoding, SoTL-E, and Jacob. cov. as input features.

**Query**(architecture $a_{\text{test}}$):

- While $t < B_{\text{query}}$, train $a_{\text{test}}$
- Compute SoTL-E using the partial learning curve, and compute Jacob. cov., and the arch. encoding of $a_{\text{test}}$
- Predict val. acc. of $a_{test}$ using $m$ and the above features.

---

Table 3: Comparison of hybrid learning curve + model-based methods on NAS-Bench-201 CIFAR-10, reporting the mean Kendall Tau rank correlation along with the standard deviation.

| Init. time (s) | Query time (s) | LC-prev-builds | LCNet | LcSVR |
|:---:|:---:|:---:|:---:|:---:|
| $1.4e4$ | 64 | $\mathbf{0.512 \pm 0.043}$ | $0.322 \pm 0.128$ | $0.427 \pm 0.181$ |
| $1.4e4$ | 238 | $\mathbf{0.676 \pm 0.029}$ | $0.204 \pm 0.209$ | $0.484 \pm 0.172$ |
| $1.4e4$ | 932 | $\mathbf{0.780 \pm 0.019}$ | $0.317 \pm 0.265$ | $0.497 \pm 0.174$ |
| $7.6e4$ | 64 | $0.516 \pm 0.042$ | $0.388 \pm 0.104$ | $\mathbf{0.596 \pm 0.065}$ |
| $7.6e4$ | 238 | $0.688 \pm 0.026$ | $0.471 \pm 0.084$ | $\mathbf{0.700 \pm 0.052}$ |
| $7.6e4$ | 932 | $\mathbf{0.797 \pm 0.016}$ | $0.583 \pm 0.090$ | $0.740 \pm 0.048$ |
| $2.2e5$ | 64 | $0.523 \pm 0.041$ | $0.418 \pm 0.074$ | $\mathbf{0.632 \pm 0.044}$ |
| $2.2e5$ | 238 | $0.692 \pm 0.023$ | $0.472 \pm 0.097$ | $\mathbf{0.736 \pm 0.034}$ |
| $2.2e5$ | 932 | $\mathbf{0.797 \pm 0.014}$ | $0.613 \pm 0.077$ | $0.795 \pm 0.023$ |
| $1.1e6$ | 64 | $0.528 \pm 0.039$ | $0.452 \pm 0.081$ | $\mathbf{0.667 \pm 0.045}$ |
| $1.1e6$ | 238 | $0.698 \pm 0.025$ | $0.472 \pm 0.101$ | $\mathbf{0.759 \pm 0.032}$ |
| $1.1e6$ | 932 | $0.798 \pm 0.014$ | $0.637 \pm 0.076$ | $\mathbf{0.823 \pm 0.023}$ |

## C    Additional Experiments on NAS-Bench-201

In this section, we give additional experiments carried out on NAS-Bench-201 CIFAR-10. These experiments include implementing additional hybrid predictors, running zero-cost proxy baselines, computing the variance in random seeds when the training and testing datasets are fixed, and running the predictors with longer HPO budgets.

### C.1    Additional hybrid predictors

In Section 4, we implemented one predictor that was a hybrid learning curve + model-based method: LcSVR [2]. In this section, we implement two other hybrid predictors: LCNet [20] and "LCE using previous builds" [8]. For both techniques, we used the original implementation.

We compute the Kendall Tau rank correlation of LcSVR, LCNet, and LC-prev-builds on a representative set of four different initialization times and three different query times, for twelve total settings on NAS-Bench-201 CIFAR-10. See Table 3. We find that LC-prev-builds outperforms LcSVR when the initialization time is small, and when the query time is high. LCNet never outperformed LcSVR.

### C.2    Zero-cost proxy baselines

In this section, we implement flops and params as baselines for zero-cost predictors. We compare the Kendall Tau rank correlation and Spearman rank correlation on NAS-Bench-201 CIFAR-10 for flops, params, and all of the zero-cost proxies implemented in Section 4. Note that zero-cost predictors

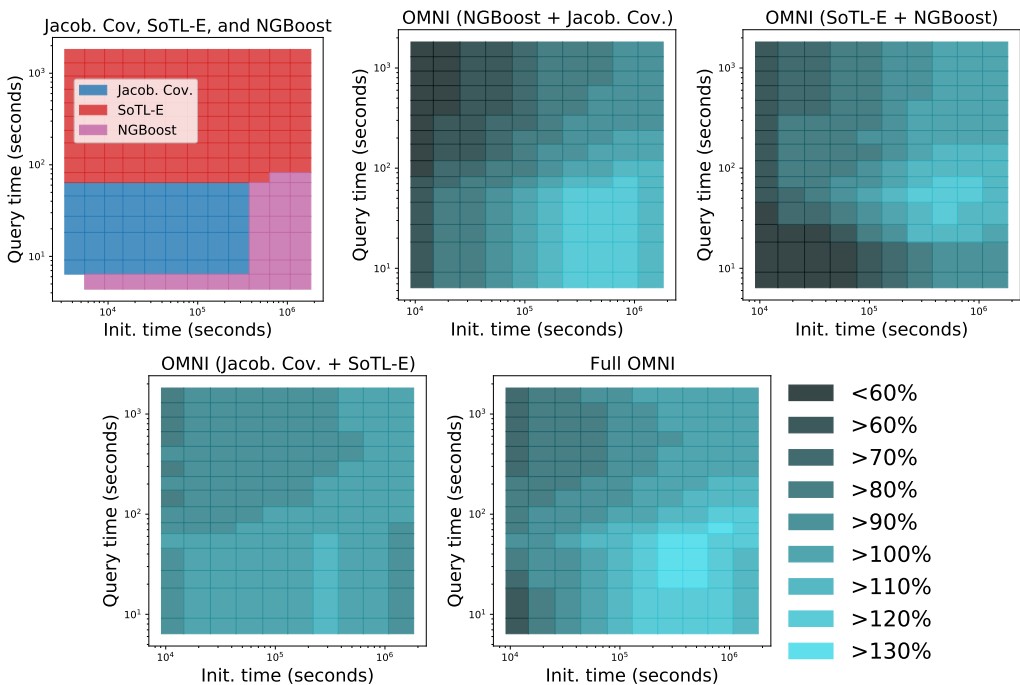

Figure 10: Ablation for OMNI. The first plot shows the best predictor in {Jacob. Cov. SoTL-E, and NGBoost} for each initialization time and query time budget. In the next four plots, we compute the percentage of the Kendall Tau value of the given OMNI variant, compared to the first plot. For example, in the bottom-right corner of the first plot, NGBoost achieves the highest Kendall Tau value from the set {Jacob. Cov. SoTL-E, and NGBoost}; In the bottom-right corner of the second plot, (NGBoost + Jacob. Cov.) achieves a Kendall Tau value that is 10% higher than the Kendall Tau value achieved by NGBoost. Therefore, combining Jacob. Cov. with NGBoost achieved a higher Kendall Tau value than the best individual predictor.

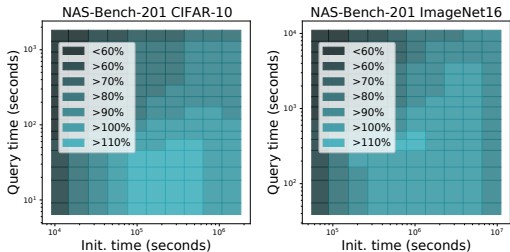

Figure 11: Percentage of OMNI (SemiNAS)'s Kendall Tau value compared to the next-best predictors for each budget constraint.

have just one setting: no initialization time and a query time of 5 seconds. See Table 4. Surprisingly, flops and params tie for the second highest Kendall Tau value out of all six zero-cost predictors on NAS-Bench-201 CIFAR-10, behind Jacobian covariance. For Spearman values, flops and params tie for third behind Jacobian covariance and SynFlow. Note that for the case of NAS-Bench-201, since the graph structure of the cell is fixed, flops has a one-to-one relationship with params (meaning they get the same rank correlation results).

## C.3 Random seed experiments

Our plots in Figure 6 give the standard deviation across 100 trials for each predictor across different random seeds, which varies the tran and test sets as well as any stochasticity of the predictor. Now, we perform another experiment where we keep the train and test sets fixed and measure the standard

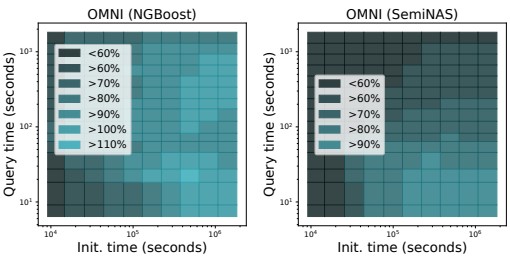

Figure 12: Percentage of OMNI's Kendall Tau value compared to the next-best predictors for each budget constraint, in the mutation-based setting on NAS-Bench-201 CIFAR-10.

Table 4: Comparison of baseline methods to zero-cost proxies on NAS-Bench-201 CIFAR-10.

| Method | Kendall Tau | Spearman |
|---|---|---|
| Flops | 0.539 | 0.713 |
| Params | 0.539 | 0.713 |
| Fisher | 0.218 | 0.299 |
| Grad Norm | 0.420 | 0.587 |
| Grasp | 0.330 | 0.481 |
| Jacob. Cov. | **0.575** | **0.743** |
| SNIP | 0.422 | 0.590 |
| SynFlow | 0.530 | 0.724 |

deviation across only the stochasticity of the predictor. For a representative set of eight predictors on NAS-Bench-201 CIFAR-10, we computed the standard deviation of 10 trials on a fixed train and test set, averaged over 50 trials of choosing new train and test sets (500 trials total per predictor). We used the median settings of initialization time and query time and reported the Kendall Tau rank correlation. See Table 5. We find that Bayesian Linear Regression has the highest stochasticity.

### C.4   Longer HPO budgets

In Section 4, all model-based predictors were run using 15 minutes of hyperparameter tuning. In this section, we test whether there is further improvement when increasing the hyperparameter tuning budget from 15 minutes to 1 hour. We run this experiment on a representative set of ten model-based predictors on NAS-Bench-201 CIFAR-10, across all eleven initialization time settings from Section 4 (the query time for all model-based predictors is fixed at one second). For each predictor, after computing the improvement to Kendall Tau rank correlation when increasing the hyperparameter tuning budget, we report the average improvement over all initialization time settings, as well as the average of the top two and average of the worst two initialization time settings. See Table 6.

There were no predictors which achieved non-negligible improvement across all initialization time settings. Similarly, none of the predictors saw non-negligible decreases in performance, although BONAS had the lowest worst two average, at -0.020, likely due to additional overfitting that happens in the 1 hour hyperparameter tuning budget. For the top two average improvements, BANANAS saw the biggest improvements, and MLP, BONAS, and GCN also saw non-negligible improvements. The rest of the performance predictors saw little to no improvement. Interestingly, the largest improvements were all from deep learning based predictors.

## D   Reproducibility Table

When giving a large-scale comparison of performance predictors, it is important that all of the methods are accurately implemented and optimized. We took a number of reasonable steps to ensure this, including *(1)* using the original implementations whenever possible, and *(2)* applying light hyperparameter tuning to all methods, to help control for the fact that different techniques received different amounts of hyperparameter tuning in their original release. The best way to ensure that all

Table 5: Standard deviation of predictors when train and test sets are fixed.

| Method | Mean | Std. Dev. | Std. Dev. w. fixed datasets |
|---|---|---|---|
| BANANAS | 0.254 | 0.050 | 0.032 |
| Bayes. Lin. Reg. | 0.291 | **0.065** | **0.115** |
| Jacob. Cov | 0.539 | 0.000 | 0.006 |
| NGBoost | 0.355 | 0.059 | 0.032 |
| SoTL-E | **0.623** | 0.032 | 0.000 |
| SynFlow | 0.529 | 0.000 | 0.001 |
| Var. Sparse GP | 0.486 | 0.054 | 0.000 |
| XGBoost | 0.390 | 0.052 | 0.031 |

Table 6: Standard deviation of predictors when train and test sets are fixed.

| Method | Avg. | Worst Two Avg. | Top Two Avg. |
|---|---|---|---|
| BANANAS | 0.012 | -0.010 | **0.059** |
| Bayes. Lin. Reg. | 0.002 | -0.006 | 0.014 |
| BOHAMIANN | 0.001 | -0.003 | 0.006 |
| BONAS | 0.008 | **-0.020** | 0.036 |
| DNGO | 0.000 | -0.008 | 0.007 |
| GCN | 0.008 | -0.016 | 0.035 |
| GP | 0.003 | -0.004 | 0.009 |
| MLP | 0.009 | -0.009 | 0.039 |
| Sparse GP | -0.003 | -0.008 | 0.006 |
| Var. Sparse GP | **0.017** | -0.002 | 0.032 |

methods are properly implemented is to reproduce the results reported by the original authors of a given technique.

In this section, we present a table to clarify for which of the 31 predictors we have succeeded in reproducing published results, and for which predictors it is not possible. For each predictor, we mark whether the original paper (or first paper to run on a NAS benchmark) had *(1)* at least one search space out of the ones we used, *(2)* at least one initialization and query time that matches one of our settings, and *(3)* at least one metric from our set of metrics. If yes to *(1)-(3)*, then we check whether we *(4)* achieved nearly the same numbers as the original paper. See Table 7.

Of the 31 predictors, 14 were released before any of the NAS-Bench search spaces had come out. Two more did not give experiments in the setting of our paper. We are able to fairly compare the remaining 15 performance predictors (up to smaller changes in the experimental setting, such as different test set sizes) with either the original paper or the first paper to give results on a NAS benchmark. Our results are close to the original results, or in some cases stronger (due to our use of HPO). All 15 are within 0.04 of the reported rank correlation value or higher.

Table 7: Reproducibility table, clarifying which predictors we have reproduced with respect to existing published results.
[1] Compared with first follow-up paper to give correlation results on a NAS benchmark. [2] Larger test set. [3] Approximated from a plot.

| Predictor | Paper | Search space | Setting | Metric | Value | Ours | Diff. |
|---|---|---|---|---|---|---|---|
| BANANAS | [69] | NAS-Bench-101 | 1000 train, 100 test | Pearson | 0.699 | 0.904 | +0.205 |
| Bayes. Lin. Reg. | [6] | Released before NAS-Bench search spaces | | | | | |
| BOHAMIANN | [57] | Released before NAS-Bench search spaces | | | | | |
| BONAS | [54] | NAS-Bench-101 | Only used size 360k train set. | | | | |
| DNGO | [56] | Released before NAS-Bench search spaces | | | | | |
| Early Stop. ValAcc | [50][1] | NAS-Bench-201 | 100 epochs, 1000 test[2] | Spearman | $0.85^3$ | 0.850 | +0.0 |
| Early Stop. ValLoss | [50][1] | NAS-Bench-201 | 100 epochs, 1000 test[2] | Spearman | $0.83^3$ | 0.839 | +0.009 |
| Fisher | [1][1] | NAS-Bench-201 | 15k test[2] | Spearman | 0.36 | 0.328 | -0.032 |
| GCN | [67] | NAS-Bench101[1] | 1000 train, 100 test | Pearson | 0.607 | 0.793 | +0.186 |
| GP | [48] | Released before NAS-Bench search spaces | | | | | |
| Grad Norm | [1] | NAS-Bench-201 | 15k test[2] | Spearman | 0.58 | 0.587 | +0.007 |
| Grasp | [1][1] | NAS-Bench-201 | 15k test[2] | Spearman | 0.48 | 0.481 | +0.001 |
| Jacob. Cov. | [1] | NAS-Bench-201 | 1000 test[2] | Pearson | 0.574 | 0.575 | +0.001 |
| LCE | [11] | Released before NAS-Bench search spaces | | | | | |
| LCE-m | [20] | Released before NAS-Bench search spaces | | | | | |
| LcSVR | [50] | NAS-Bench-201[1] | 100 epochs, 100 train, 1000 test[2] | Spearman | $0.93^3$ | 0.931 | +0.001 |
| LGBoost | [18] | NAS-Bench101 | 1000 train, 100 test | Kendall Tau | 0.640 | 0.705 | +0.065 |
| MLP | [69][1] | NAS-Bench101 | 1000 train, 100 test | Pearson | 0.400 | 0.447 | +0.047 |
| NAO | [35] | Does not have NAS-Bench search spaces | | | | | |
| NGBoost | [14] | Released before NAS-Bench search spaces | | | | | |
| OneShot | [73] | Released before NAS-Bench search spaces | | | | | |
| Rand. Forest | [29] | Released before NAS-Bench search spaces | | | | | |
| RSWS | [26] | Released before NAS-Bench search spaces | | | | | |
| SemiNAS | [34] | NAS-Bench-101 | No predictor experiments | | | | |
| SNIP | [1][1] | NAS-Bench-201 | 15k test[2] | Spearman | 0.58 | 0.590 | +0.01 |
| SoTL | [50] | NAS-Bench-201 | 100 epochs, 1000 test[2] | Spearman | $0.96^3$ | 0.963 | +0.003 |
| SoTL-E | [50] | NAS-Bench-201 | 100 epochs, 1000 test[2] | Spearman | $0.93^3$ | 0.932 | +0.002 |
| Sparse GP | [46] | Released before NAS-Bench search spaces | | | | | |
| SynFlow | [1] | NAS-Bench-201 | 15k test[2] | Spearman | 0.74 | 0.738 | -0.002 |
| Var. Sparse GP | [63] | Released before NAS-Bench search spaces | | | | | |
| XGBoost | [9] | Released before NAS-Bench search spaces | | | | | |