# OpenReview forum: "How Powerful are Performance Predictors in Neural Architecture Search?"
_NeurIPS.cc/2021/Conference — NeurIPS 2021 Poster_

### Official Review · Reviewer_eo6y · 2021-06-25

**Rating:** 7
**Confidence:** 4

**Summary:**

The authors identify and benchmark a plethora of recently proposed techniques for predicting neural network architecture performance on a task without doing a full training run.

They begin by grouping each technique into categories: model-based, learning-curve-based, zero-cost, and weight-sharing. When viewed like this it becomes clear that there is a need for a common benchmark to distinguish the performance of the different approaches.

To compare under one framework, evaluation is done with respect to two variables: initialization time (time spent preparing the model) and query time (time spent retrieving predictions from the model).

The authors then present their own method, OMNI, which performs NAS using a blend of model-based, learning-curve-based, and zero-cost predictors.


**Limitations And Societal Impact:**

I have no concerns about societal impact beyond any other work on neural networks.



**Main Review:**

I think this is a solid paper and should be accepted. There is nothing hugely surprising or significant in it, and there are no real deep insights uncovered, so I'm not sure it merits being a "seminal paper". However, the experiments are thorough, logically conducted and answer previously unanswered questions. I think overall this work will be helpful and insightful to the neural architecture search (NAS) subfield.

**Originality:** This is a novel investigation into (and combination of) pre-existing techniques. Analysing different predictors under a single framework (via query/init times) is a new way of looking at NAS and allows us to ask interesting questions.  Related work is adequately cited.

**Quality:** The submission is technically sound, and the reader leaves the paper with the impression that each claim has been thoughtfully evaluated. The choice of methods seems to give a reasonable coverage of the space of existing work and the authors are careful to address problems with comparative analysis (especially with regard to hyperparameter setup).

**Clarity:** The submission is very clearly written and organized. It's easy to glean the main takeaways and the logical flow of the paper makes sense. I feel confident that there is enough detail in the paper that would allow me to reproduce each of the results.

**Significance:** The results of the paper come in the form of clear takeaways which I'm sure will be of practical use to anyone hoping to use neural architecture search. The results from the experiments are a unique dataset which will allow for the exploration of further lines of research.

## Positive points:
- The paper is very well written and structured, and comes with clear, robustly verified takeaways
- Open sourced code
- This is an experiment-heavy paper, which I suppose you would expect from a characterization task like this. The experiments cover a wide range of datasets and techniques, and at the end the authors present a combined method which appears to robustly outperform using a single approach.
- There are many interesting visualisations with pleasantly surprising results — the fact that the benefits of using different predictors can be combined and are complementary is both interesting and useful as an insight.

## Points to address:
1. I found the use of the terms: initialization, query and update a little confusing. In the case of a zero-cost proxy, does “initialization time” ever change? Whereas for something like XGBoost, does “initialization” really mean “training”?
2. When do we explicitly want to trade off query and initialization time? At the end of section 4 the authors state “we recommend deciding on a target initialization time and query time budget”. I understand that the two aspects were needed to design a comparison framework for various methods, but I’m struggling to think of a scenario in which this is a trade-off that’s useful to be able to make, or what might motivate me to have a preference for one or the other?
3. It’s an enormous topic, but I would have liked to see a slightly more fundamental investigation of what drives the various performance predictors. Is there anything that can be extracted from the best performing methods on various NAS-benchmarks to address what exactly each one is measuring? Since the different techniques appear to be complementary, is it likely that they are measuring different things? I suspect this question is out of scope for this paper (which is already packed with insights), but it would be great to have some quantification of this.
4. I would also have liked to see some more “baseline” methods. For instance, what if you use number of parameters/flops as a zero-cost proxy?
5. Minor visual note: the plot titles in figures 3 and 5 aren’t aligned properly.

## General questions

I think this is a really interesting piece of work that leaves the reader with many things to ponder! I therefore have some more general questions for the authors, that I don’t think necessarily need to be addressed in the paper but would be interested to hear about (I will continue numbering them in case you would like to reply to them specifically):

6. Regarding NAS-Bench-201, the authors state "6 466 are unique after removing isomorphisms” — do they have a citation for this?
7. On line 279 the authors state: “The NAS-Bench-101 API only gives the validation accuracy at four epochs”. Is this true? This seems surprising!
8. The authors mention that the training and test sets are architectures drawn uniformly at random from the search space — could they confirm that these are two separate sets? I.e. there’s no chance of resampling the same architecture?
9. During the mutation-based test set experiments the authors limit the test architectures to be “at most an edit distance of three from a seed architecture”. How is edit distance calculated? Is it a distance between binary adjacency matrices? If so, is the edit distance of replacing e.g. a conv1x1 with a conv3x3 the same as the edit distance of replacing a conv3x3 with an identity? Again, it’s not necessary to include anything in the paper, but I would be intrigued to hear the authors thoughts on how different measurements of “distance” might impact this set of experiments. Can we use e.g. the distance between scalar outputs of zero-cost measures as “edit distances” between architectures?




**Time Spent Reviewing:**

4

---

> ### Author Response · Authors · 2021-08-10
> **Thank you; we have added your suggestions**
>
> We thank the reviewer for their insightful and positive review, and for all of the comments and suggestions that can further improve our paper. We reply to all comments below.
>
> 1. *Initialization, query and update are confusing.* We agree that these definitions are confusing - it was our attempt to have a very general framework for all types of predictors. For model-based predictors, initialization time is the time it takes to (a) train the architectures which make up the training set, and (b) train the performance predictor model using this training set. For zero-cost predictors, the only required computation is specific to each queried architecture, which is classified under query time, so the initialization time is zero. We will clarify this in the paper, and we are open to any suggestions for better names than initialization/query/update.
>
> 2. *When do we trade off query and initialization time?* This is a great question - it depends on a few factors such as the type of NAS algorithm and the total runtime budget, and different settings are needed in different situations. For example, if there are many architectures whose performance we want to estimate, then we should have a low query time, and if we have a high total runtime budget, then we can afford a high initialization time. We may also change our runtime budget throughout the run of a single NAS algorithm. For example, at the start of a NAS algorithm, we may want to have coarse estimates of a large number of architectures (low initialization time, low query time such as zero-cost predictors). As the NAS algorithm progresses, it is more desirable to receive higher-fidelity predictions on a smaller set of architectures (model-based or hybrid predictors). The exact budgets depend on the type of NAS algorithm. We will add this discussion into the paper.
>
> 3. *Investigation of what drives the predictors.* This is an interesting suggestion, and we agree that it warrants more discussion in the paper. From the OMNI ablation study in Figure 10 and our knowledge of how the predictors work, we can hypothesize that each predictor type measures distinct quantities: SOTL measures the training speed, zero-cost predictors measure the covariance between activations on different datapoints, and model-based predictors simply learn patterns between the architecture encodings and the validation accuracies. A full quantification of the correlation between these measurements is a great question for future work.
>
> 4. *More baseline methods.* Thanks for the suggestion. We added flops and params as baselines for zero-cost predictors. Surprisingly, flops and params tie for the 2nd highest Kendall Tau value out of all six zero-cost predictors on NAS-Bench-201 cifar10, behind Jacobian covariance. For Spearman values, flops and params tie for 3rd behind Jacobian covariance and SynFlow.
>
> 5. *Align plot titles.* Thanks, we will properly align the titles in Figures 3 and 5.
>
> 6. *Claim that NAS-Bench-201 has 6466 architectures after removing isomorphisms.* Yes, the original [NAS-Bench-201 paper](https://arxiv.org/pdf/2001.00326.pdf) mentions this in the first paragraph of Appendix A. We will add this citation to the paper.
>
> 7. *Claim that NAS-Bench-101 only gives accuracies at four epochs.* Yes, NAS-Bench-101 only gives the validation accuracies for epochs 4, 12, 36, and 108. From personal correspondence with the authors, we verified that they unfortunately did not save the accuracies at the rest of the epochs between 1 and 107.
>
> 8. *Confirm that train and test are disjoint sets.* Yes, in all of our experiments we removed all overlaps between the training and test sets (mentioned on line 304).
>
> 9. *How is edit distance calculated?* Throughout our paper, we went with the simplest and most standard definition of edit distance from the NAS literature: two architectures are a single edit distance away if they differ by one operation or edge. The question of defining edit distance is related to other interesting recent work in NAS such as encoding optimization [1,2], and search space optimization [3]. We think the reviewer’s idea of using zero-cost measures to define edit distance is very interesting!
>
> We thank the reviewer for their helpful comments. If you have any more questions or follow-up comments, please let us know.
>
> [1] Does Unsupervised Architecture Representation Learning Help Neural Architecture Search?, Yan et al.
> [2] A Study on Encodings for Neural Architecture Search, White et al.
> [3] Neural Architecture Generator Optimization, Ru et al.

---

### Official Review · Reviewer_RXE7 · 2021-07-12

**Rating:** 7
**Confidence:** 4

**Summary:**

Neural Architecture Search (NAS) aims to find the network architectures with the best accuracies -- or best size/accuracy tradeoffs -- from within a human-defined search space. The submission evaluates a number of different heuristics for ranking different candidate architectures in a search space. The submission claims to evaluate 31 different techniques which fall into several categories (e.g., early stopping, learning curve extrapolation, supervised learning, one-shot models, zero-shot metrics). The submission claims that different techniques work better for different compute budgets (e.g., low setup cost/low search cost vs. high setup cost/high search cost). It also claims that combining techniques from different families (e.g., supervised learning + zero-shot metrics) can lead to better results than using any one technique on its own. Experimental results are reported on NASBench-101, NASBench-201, the NASBench-301/DARTS search space, and NASBench-NLP.

**Limitations And Societal Impact:**

The quality of ML algorithms tends to be sensitive to hyper-parameters and small implementation details, so the reliability of the submission's claims may depend on the quality of the submission's authors' implementations of different NAS techniques. However, this is as much a general concern about the type of work (trying to evaluate/compare/rank 30+ techniques in a single paper) than about authors' specific implementation.

The most effective mitigation I can think of is to verify that the submission's implementation is able to reproduce the results reported by the original authors of a given technique. I don't think the submission's authors are doing this, but I might've missed something. (I haven't checked the paper's appendix too closely.)

**Main Review:**

**Contributions:** The submission evaluates/compares a number of previously proposed NAS techniques from several popular categories:
1. Supervised training methods which sample a set of (random) architectures, measure the performance (accuracy) of each one, and then use the resulting  `<architecture, accuracy>` pairs to train a regression model which can predict the accuracies of unseen architectures.
2. Curve extrapolation ("learning-curved based") methods which try to predict a network's accuracy at the end of training by measuring its accuracy at each of the first `K` epochs of training.
3. One-shot NAS methods which train a single set of shared weights which can be used to evaluate/rank many different candidate architectures in the search space.
4. Zero-shot NAS methods which try to rank different candidate architectures based on a few batches of data using randomly initialized weights.

**Results:** The submission (not surprisingly) claims that different techniques are optimal in different ranges. For example, it claims that Jacobian Covariance, a zero-shot technique, performs best in the very low-cost regime, whereas simply training network architectures from scratch for a shortened number of steps performs best in the high-cost regime. Both of these claims seem plausible.

Perhaps more interesting -- and more surprising -- is the submission's claim that if the goal is to maximize the correlation between predicted and observed (ground-truth) model accuracies, "Of the 31 predictors we tested, we found that just seven of them are Pareto-optimal with respect to Kendall Tau, initialization time, and query time. That is, only seven algorithms have the highest Kendall Tau value for at least one of the 154 query time/initialization time budgets on NAS-Bench-201 CIFAR-10" (L217). Similar trends are reported for the other datasets.

In addition to evaluating different techniques' abilities to rank different candidate architectures in a search space (e.g., rank-correlation between predicted and ground-truth accuracies), the submission also evaluates: (i) each technique's ability to estimate which of two similar-looking architectures will have a higher accuracy, and (ii) the effects of ensembling different techniques, e.g., a supervised technique + zero-shot NAS.

**Significance:** The findings, if valid, can be potentially useful for practitioners who want to know which classes of NAS techniques to focus on. One risk in trying to evaluate so many different techniques in a single paper is that the authors might not spend enough time ensuring that each individual technique is properly implemented and well-optimized. The authors take a number of reasonable steps to try to mitigate this concern:
1. They try to use the authors' original implementations of different techniques in many cases (L186). However, looking at the Appendix and included source code, I have the impression that some techniques are evaluated using snippets of code from the original authors' implementation whereas others are reimplemented from scratch.
2. Hyper-parameters are re-tuned by the submission's authors to help control for the fact that different techniques probably received different amounts of hyper-parameter tuning in the past.
3. The final evaluation uses a separate held-out test set. (Good ML practice.)

Given the sheer number of architectures evaluated by the submission, this seems like a reasonable evaluation protocol. However, it comes with the caveat that if a technique is found *not* to work well by the submission's authors, it can be unclear whether that's because of a flaw in the technique itself or whether the submission's implementation of that technique is implemented incorrectly / insufficiently tuned.

The submission's claim that ensembling different NAS techniques (e.g., NGBoost + Jacobian covariance) can lead to significantly improved performance seems intuitive and potentially relevant for practitioners. This seems like a direction that's received relatively little attention for the NAS community. While it comes with the downside of making the NAS implementation significantly more complex, I still think reporting these results is a nontrivial research contribution.

**Clarity:** The submission seems well-written, polished, and easy to understand.

**Post-rebuttal update:** The discussion and reproducibility table provided by the authors does a good job of addressing my concerns about whether the original methods were reproduced faithfully, and I've increased my rating from 6 to 7.

**Time Spent Reviewing:**

5

---

> ### Author Response · Authors · 2021-08-10
> **We agree; we have further validated the performance of all predictors**
>
> We thank the reviewer for their insightful comments and favorable review. We agree with the concern that since we have implemented 31 predictors, some of them may not have been fully tuned. Although we did our best to use the official code when possible, and run HPO, this still could be a concern for some predictors. We made two additions to our paper to address this.
>
> First, we added a table to clarify for which of the 31 predictors we have succeeded in reproducing the original results, and for which predictors it is not possible. For each predictor, we mark whether the original paper had (1) at least one search space out of the ones we used, (2) at least one initialization and query time that matches one of our settings, and (3) at least one metric from our set of metrics. If yes to (1)-(3), then we check whether we (4) achieved nearly the same numbers as the original paper. We post the table below.
>
> Of the 31 predictors, 14 were released before any of the NAS-Bench search spaces had come out. Three more did not give experiments in the setting of our paper. We were able to fairly compare the remaining 14 performance predictors (up to smaller changes in the experimental setting, such as different test set sizes) with either the original paper or the first follow-up paper to give results on a NAS-Bench search space. Our results were close to the original results, or in some cases stronger (due to our use of HPO). All 14 were within about 0.1 of the reported rank correlation value or higher, and 12 of the 14 were within 0.02 or higher.
>
> Second, we ran a new experiment that increases the HPO budget from 15 minutes to 1 hour, and we tracked which predictors further improved. BANANAS saw the biggest improvements, increasing Kendall Tau for the two largest initialization time settings by 0.059. MLP also saw non-negligible improvements in the largest initialization time settings, by 0.039. BONAS and GCN also had smaller improvements (0.036) in the medium initialization time settings. The rest of the performance predictors saw little to no improvement. Interestingly, the largest improvements were all from deep learning based predictors.
>
> We thank the reviewer again as this has led to improved verification of the results in our paper. Please let us know if you have any further concerns or clarifications, and if you are satisfied with our response, we respectfully ask that you consider increasing your score.

---

> > ### Author Response · Authors · 2021-08-10
> > **Reproducibility table**
> >
> > For each predictor, we check whether
> > (1) at least one search space out of the ones we used,
> > (2) at least one initialization and query time that matches one of our settings, and
> > (3) at least one metric from our set of metrics. If yes to 1-3, then
> > (4) we check whether we achieved nearly the same numbers as the original paper.
> >
> > Note that our improvement over the original BANANAS is consistent with our observation above that it sees substantial improvements with HPO.
> >
> > |Predictor|Search space|Setting|Metric|Value|Ours|Diff.|
> > |------------|----------------|--------|------|-----|----|-----|
> > |BANANAS|NB101|1000 train, 100 test|Pearson|0.699|0.904|+0.205|
> > |Bayes Lin Reg|Released before NB search spaces|
> > |BOHAMIANN|Released before NB search spaces|
> > |BONAS|NB101|360k train, 21k test|
> > |DNGO|Released before NB search spaces|
> > |Early Stop. ValAcc|NB201$^1$|100 epochs, 1000 test$^2$|Spearman|0.85$^3$|0.850|+0.0|
> > |Early Stop. ValLoss|NB201$^1$|100 epochs, 1000 test$^2$|Spearman|0.83$^3$|0.839|+0.009|
> > |Fisher|NB201|15k test$^2$|Spearman|0.36|0.328|-0.032|
> > |GP|Released before NB search spaces|
> > |GCN|NB101$^1$|1000 train, 100 test|Pearson|0.607|0.793|+0.186|
> > |Grad Norm|NB201|15k test$^2$|Spearman|0.58|0.587|+0.007|
> > |Grasp|NB201|15k test$^2$|Spearman|0.48|0.481|+0.001|
> > |Jacob. Cov.|NB201|1000 test$^2$|Pearson|0.574|0.575|+0.001|
> > |LCE|Released before NB search spaces|
> > |LCE-m|Released before NB search spaces|
> > |LcSVR|NB201$^1$|100 epochs, 100 train, 1000 test$^2$|Spearman|0.93$^3$|0.931|+0.001|
> > LGBoost|NB101|1000 train, 100 test|Kendall Tau|0.640|0.705|+0.065|
> > |MLP|NB101|1000 train, 100 test|Pearson|0.400|0.447|+0.047|
> > |NAO|Does not have NAS-Bench search spaces (released just after)|
> > |NGBoost|Released before NB search spaces|
> > |OneShot|Released before NB search spaces|
> > |Rand. Forest|Released before NB search spaces|
> > |RSWS|Released before NB search spaces|
> > |SemiNAS|NB101|No predictor experiments|
> > |SNIP|NB201|15k test$^2$|Spearman|0.58|0.590|+0.01|
> > |SoTL|NB201|100 epochs, 1000 test$^2$|Spearman|0.96$^3$|0.963|+0.003|
> > |SoTL-E|NB201|100 epochs, 1000 test$^2$|Spearman|0.93$^3$|0.932|+0.002|
> > |Sparse GP|Released before NB search spaces|
> > |SynFlow|NB201|15k test$^2$|Spearman|0.74|0.738|-0.002|
> > |Var. Sparse GP|Released before NB search spaces|
> > |XGBoost|Released before NB search spaces|
> >
> > $^1$ Compared with first follow-up paper to give correlation results on a NAS-Bench search space.
> > $^2$ Larger test set.
> > $^3$ Approximated from a plot.

---

> > > ### Comment · Reviewer_RXE7 · 2021-08-19
> > > **Thanks for adding reproducibility results**
> > >
> > > I think your explanation and reproducibility table do a good job of addressing my concerns about whether the original methods were implemented correctly, and I'll revise my score accordingly.

---

> ### Author Response · Authors · 2021-08-19
> **Update on the reproducibility table**
>
> Hi, we made a few changes to the reproducibility table below. We successfully reproduced one additional result and fixed the reason for slight discrepancies in three more. We updated the table below, and we note the changes here:
> - *LGBoost:* the authors only reported results for their own hand-designed metric ([end of Section 5.1](https://arxiv.org/pdf/2007.04785.pdf)). However, upon closer inspection, we found that their metric is exactly equal to $2*(\text{Kendall Tau}) - 1$ [1]. Using this formula to make a fair comparison, we found that we slightly outperformed the result listed in their paper.
> - *Early Stop Val Loss:* Originally, this was our biggest discrepancy at -0.09. We realized that the result listed in the paper summed up validation losses from all epochs, instead of returning the validation loss at the final epoch. With this correction, the difference between our result and the reported result became +0.009. We also plan to update the results in our paper, since this method outperforms Early Stop Val Loss.
> - *Synflow:* Originally, we reported a discrepancy of -0.016 with an asterisk that we used a smaller test set. We re-ran with a larger test set and the difference became -0.002. This is also consistent with our [response to reviewer Z9bu](https://openreview.net/forum?id=6RB77-6-_oI&noteId=c-0rneHkZ56) about test set sizes.
> - *Fisher:* Similar to synflow, we re-ran with a larger test set and decreased the difference to -0.032.
>
> Overall, we have now fairly compared 15 predictors to their original paper, and all 15 are within 0.04 of the original result or higher. The original papers for the remaining predictors either were released earlier than NAS benchmarks, or did not contain rank correlation experiments.
>
> [1] This follows from the [definition of Kendall Tau](https://en.wikipedia.org/wiki/Kendall_rank_correlation_coefficient), and the fact that #concordant + #discordant = ${n \choose 2}$.

---

### Official Review · Reviewer_Z9bu · 2021-07-18

**Rating:** 7
**Confidence:** 5

**Summary:**

This paper compares 3 kinds of performance predictors, model based predictors, learning curve based predictor and zero nas predictors. They benchmark the performance of all 31 predictors against 5 benchmarks: NAS-Bench101, DARTS, NASBench201 and NLP.

In addition to this, they also incorporate SOTL-E and jacobian variance as addition features to NGBOOST and SemiNAS predictors and demonstrate that it improves the performance over all 3 of them.

**Limitations And Societal Impact:**

1) While SoTL and SoTL-E perform well empirically, it is theoretically not valid to use sum of training losses to predict the performance of a network.  A network with lower training loss might also be overfitting and not generalize, while a network with regularization such as dropout, l2 etc might have a higher training loss but still generalize well.

2) It might be better if you didn't include SoTL and SoTL-E in your list of 31 predictors as well.

3) Given that the we know the learning curves of all the architectures in the NAS benchmarks, why not include all the architectures in the test set, rather than only 200? That will give us an actual overview of how these predictors perform on that search space.

4) Given that OMNI has a SEMINAS variant as well, please generate figures corresponding to Figure 3 and 10 to quantify how much it improves over SEMINAS too.

5) Please include OMNI in figure 4 to show its efficacy in ranking architectures that are close to each other.

6) Another useful experiment to perform would be to test how stable the predictors are to various train/test splits and random seeds.

7) I understand that the predictors included in the paper are all those which have open source repositories. It is useful to include other state of the art (SOTA) learning curve predictors ([1] ; [2]; 3) and SOTA model based predictors [4] to a more complete analysis of the performance predictors.

[1] Learning curve prediction with Bayesian neural networks, Klein et al.
[2] Learning to Rank Learning Curves, Wistuba et al.
[3] Speeding up hyperparameter optimization by extrapolation of learning curves using previous builds, Chandrasekhar et al.
[4] Neural Architecture Search in A Proxy Validation Loss Landscape, Li et al.

**Main Review:**

Comparing all the predictors and analyzing about how they perform with respect to initialization time and query time is definitely useful for the community.

They were able to empirically demonstrate some hunches that the community had such as 1) Zero cost NAS techniques do not perform well on larger DARTS space 2) Surrogate models using latent features of architectures would generalize well when one does not have a lot of training data.

While the experiments and the insights provided in the paper are useful, none of it is a revelation or surprising. For instance, the tree based methods developed recently are able to outperform model based methods but it takes a lot longer to train them. This paper does not have enough novelty to be accepted as a conference paper. However, this is a good starting point for people who are new to the field. It might be good to expand this paper by adding more algorithmic details of various flavors of performance predictors and make it a survey paper instead.





**Time Spent Reviewing:**

4

---

> ### Author Response · Authors · 2021-08-10
> **Thank you; we are working on all of your suggestions**
>
> We thank the reviewer for their very thorough review and helpful comments. We are pleased that the reviewer agrees that this work is beneficial to the community.
>
> While we agree with the reviewer that our paper has lower novelty than other types of papers, we respectfully point out that experimental survey-type papers have recently been accepted to ICLR [1,2] and ICML [3]. The reviewer may already be familiar with these discussions, but just in case they want a refresher, two of the meta-reviews are available [here](https://openreview.net/forum?id=HygrdpVKvr&noteId=C5rPJ2sLzP) and [here](https://openreview.net/forum?id=SJgIPJBFvH&noteId=8EatWJ_2_U).
>
> (1 & 2) *SoTL and SoTL-E are not valid.* We agree that SoTL and SoTL-E are not appropriate for joint HPO+NAS search spaces, when dropout and l2 regularization can vary. This is also true for other predictors, such as the zero-cost predictors. However, standard NAS search spaces such as NAS-Bench-101/201/301 keep all training parameters fixed. SoTL, SoTL-E, and zero-cost predictors are being used in the NAS community for performance prediction (and [2] gives theoretical motivation for SoTL for NAS), and so we will add a clear warning in our paper that SoTL, SoTL-E, and zero-cost predictors cannot be used on joint HPO+NAS search spaces.
>
> (3) *Use all architectures in the test set.* We agree with this suggestion and we are starting to re-run some of our main experiments with the full remaining set of architectures. Since Figures 2-4 compare predictors trained with different amounts of training data, we will subtract the maximum training set size to ensure there is a fair comparison and no overlap between the train and test sets. Our tests so far have shown little difference when using a much larger test set, potentially because even though we used a test set of size 200, we averaged over 100 trials for each experiment.
>
> (4) *Run OMNI-SemiNAS in the setting of Figure 3.* Thank you for the suggestion. We ran OMNI-SemiNAS, and we saw that similar to OMNI-NGB, it far outperforms its individual components, improving over SemiNAS by 19%. We also found that OMNI-SemiNAS overall performs slightly worse than OMNI-NGB (Kendall Tau decreases by 0.0891 on average over all runtime settings on NAS-Bench-201 cifar10).
>
> (5) *Run OMNI in the setting of Figure 4.* Thank you for the suggestion. We ran OMNI-NGB and OMNI-SemiNAS in the mutation-based sampling setting. Overall, they behave similarly as they did in the random sampling setting, although comparatively a bit worse, since it is a more challenging setting. OMNI-NGB achieves a top improvement of 7% Kendall Tau over the second-best predictor on NAS-Bench-201 cifar10.
>
> (6) *Stability of predictors to train/test splits and random seeds.* Our plots in the supplementary material (Figure 6) do give the variance for predictors with different random train and test sets across 100 trials. We like your idea of doing another experiment where we keep the train and test sets fixed and measure the variance across random seeds, so we will run this experiment as well.
>
> (7) *Include four more predictors.* Thank you for these suggestions. Unfortunately, 3 of the 4 will take non-trivial time to include. The Chandrasekhar et al. predictor is implemented in mxnet, and (as the reviewer points out) both Li et al. and Wistuba et al. do not have publicly available code. We are working on adding Chandrasekhar et al. However, we were able to add one of your suggestions: LCNet from Klein et al. It is a hybrid model- and learning curve-based predictor. Compared to the other hybrid predictor, LcSVR, it performs similarly or slightly worse in most settings, although it performs substantially worse in the setting of high initialization, low query time. We think this is because the learning curve extrapolation model in LCNet cannot handle the noise in the early epochs of NAS-Bench-201 learning curves.
>
> If you are satisfied with our response, we respectfully ask that you consider increasing your score. Also, please let us know any further concerns or comments. We are happy to reply.
>
> [1] NAS evaluation is frustratingly hard, Yang et al., 2020.
> [2] Fantastic Generalization Measures and Where to Find Them, Jiang et al., 2020.
> [3] Descending through a Crowded Valley - Benchmarking Deep Learning Optimizers, Schmidt et al., 2021.
> [4] Speedy Performance Estimation for Neural Architecture Search, Ru et al., 2020.

---

> > ### Author Response · Authors · 2021-08-14
> > **We have now addressed all of your suggestions**
> >
> > We have now finished the rest of your suggestions. Thank you again for providing a thorough review with helpful suggestions.
> >
> > (7) *Include four more predictors.* We implemented the technique from "Speeding up hyperparameter optimization by extrapolation of learning curves using previous builds", Chandrasekhar et al. Together with LcSVR, and your other suggestion LC-Net, we have implemented three hybrid LCE + model-based predictors. Here is a table that shows the representative performance for different initialization time and query time settings, reporting Kendall Tau:
> >
> > |Init. time (s)|Query time (s)|Chandrasekhar et al.|LCNet|LcSVR|
> > |--------------|--------------|-------------|-------------|-------------|
> > |$1.4e4$|$64$|$0.512\pm0.043$|$0.322\pm0.128$|$0.427\pm0.181$|
> > |$1.4e4$|$238$|$0.676\pm0.029$|$0.204\pm0.209$|$0.484\pm0.172$|
> > |$1.4e4$|$932$|$0.78\pm0.019$|$0.317\pm0.265$|$0.497\pm0.174$|
> > |$7.6e4$|$64$|$0.516\pm0.042$|$0.388\pm0.104$|$0.596\pm0.065$|
> > |$7.6e4$|$238$|$0.688\pm0.026$|$0.471\pm0.084$|$0.7\pm0.052$|
> > |$7.6e4$|$932$|$0.797\pm0.016$|$0.583\pm0.09$|$0.74\pm0.048$|
> > |$2.2e5$|$64$|$0.523\pm0.041$|$0.418\pm0.074$|$0.632\pm0.044$|
> > |$2.2e5$|$238$|$0.692\pm0.023$|$0.472\pm0.097$|$0.736\pm0.034$|
> > |$2.2e5$|$932$|$0.797\pm0.014$|$0.613\pm0.077$|$0.795\pm0.023$|
> > |$1.1e6$|$64$|$0.528\pm0.039$|$0.452\pm0.081$|$0.667\pm0.045$|
> > |$1.1e6$|$238$|$0.698\pm0.025$|$0.472\pm0.101$|$0.759\pm0.032$|
> > |$1.1e6$|$932$|$0.798\pm0.014$|$0.637\pm0.076$|$0.823\pm0.023$|
> >
> > For the other two predictors you suggested, we emailed the first authors to see if they could either make their code public or share it with us. Using the original code would be the best case in terms of reproducibility. We have not received a response so far.
> >
> > (3) *Use all architectures in the test set.* We ran experiments to check the difference between small and large test sets. In each case, the difference between a test set of size 200 and 14000 is within one standard deviation. We believe this is because for both settings, we averaged over at least 50 trials. Here is a representative set of results, using the median settings of initialization time and query time, reporting Kendall Tau:
> >
> > |Predictor|Test set 200|Test set 14000|Diff.|
> > |-------------|-----------|-----------|-------|
> > |BANANAS|$0.255\pm0.068$|$0.249\pm0.048$|$0.006$|
> > |Bayes. Lin. Reg.|$0.308\pm0.154$|$0.323\pm0.138$|$-0.014$|
> > |Rand. Forest|$0.437\pm0.061$|$0.426\pm0.045$|$0.011$|
> > |SoTL-E|$0.737\pm0.025$|$0.736\pm0.005$|$0.001$|
> > |Var. Sparse GP|$0.493\pm0.063$|$0.502\pm0.04$|$-0.009$|
> > |XGBoost|$0.393\pm0.061$|$0.4\pm0.045$|$-0.007$|
> >
> >
> > (6) *Stability of predictors to train/test splits and random seeds.* Thanks, this is an interesting and valuable experiment which can show which predictors are highly stochastic even when the train and test sets are fixed. For each predictor, we computed the standard deviation of 10 trials on a fixed train and test set, averaged over 50 trials of choosing new train and test sets (500 trials total per predictor). Here is a representative set of results, using the median settings of initialization time and query time, reporting Kendall Tau:
> >
> > |Predictor|mean|stdev|stdev with train/test sets fixed|
> > |-------------|------|------|---------------------------------|
> > |BANANAS|0.254|0.05|0.032|
> > |Bayes. Lin. Reg.|0.291|0.065|0.115|
> > |Jacob. Cov|0.539|0.0|0.006|
> > |NGBoost|0.355|0.059|0.032|
> > |SoTL-E|0.623|0.032|0.0|
> > |SynFlow|0.529|0.0|0.001|
> > |Var. Sparse GP|0.486|0.054|0.0|
> > |XGBoost|0.39|0.052|0.031|
> >
> > If you have any follow-up questions or concerns, please let us know. If we have addressed your concerns we would very much appreciate it if you could consider increasing your score.

---

> > ### Comment · Reviewer_Z9bu · 2021-08-19
> > **Thanks for addressing my comments**
> >
> > You are right. I apologize for overlooking the fact that experimental papers have been accepted.
> > You have addressed all my concerns. Thanks for including additional predictors, testing on the entire test set for NASBench datasets and looking at the impact of random seed. I am increasing my score.

---

### Official Review · Reviewer_sow2 · 2021-07-19

**Rating:** 7
**Confidence:** 3

**Summary:**

The paper give a large-scale comparison of performance predictors in NAS. The results act as recommendations for the best predictors to use in different settings. The author also show that certain families of predictors can be combined to achieve even better predictive power.

**Limitations And Societal Impact:**

(a) The paper has well-study the ranking performance of predictors, however I believe in NAS, sampling (of arch and peformmance pairs) also plays a important role and can largely affect the final performance of NAS predictors. As a result I believe sampling should be decoupled in the evaluation of predictor-based NAS, e.g. BANANNAS/ BONAS which use guided sampling method, should fall into a different category and compared seperately.

**Main Review:**

(a) The paper proposed a large-scale comprehensive comparison of performance predictors in NAS, the choice of predictors range from learning curve extrapolation, to weight-sharing, to 10 supervised learning and zero-cost proxies. The survey itself is beneficial to the NAS community.
(b) The paper proposed a new predictor OMNI, to combine complementary information from three families of performance predictors, leading to substantially improved performance.

**Time Spent Reviewing:**

1

---

> ### Author Response · Authors · 2021-08-10
> **Thank you for your review**
>
> We thank the reviewer for their favorable review and for recognizing that our paper is beneficial to the NAS community.
>
> The reviewer mentioned that sampling plays an important role in NAS, and we agree. Many of the model-based performance predictors in our paper are part of algorithms that use a guided sampling technique such as Bayesian optimization. In our paper, we tested the predictors in three separate ways: (1) predictor-only comparison using uniform sampling (Figure 2), (2) predictor-only comparison using mutation-based sampling that resembles the distribution in BANANAS/BONAS/etc, (Figure 4), and (3) NAS comparison using two different NAS frameworks (Figure 5). Experiment (3) especially shows which predictors are best when guided sampling is taken into account.
>
> The reviewer brings up a great point that we can run even more experiments that test which predictors are best at guided search. To address this, we conducted a new experiment in which we test the performance of the model-based predictors at three different stages of the NAS algorithm (20, 60, and 100 iterations into Bayesian optimization). At each stage, we compute each predictor’s ability to rank architectures it encounters during NAS, using Kendall Tau rank correlation (KT). We evaluated BANANAS, BONAS, GCN, MLP, NAO, XGBoost, and NGBoost on NAS-Bench-201 cifar10.
>
> |Iterations|20|60|100|
> |----------|--------|--------|--------|
> |BANANAS|0.075|0.164|0.224|
> |BONAS|0.304|0.334|0.349|
> |GCN|0.189|0.396|0.465|
> |MLP|0.127|0.262|0.316|
> |NAO|0.351|0.452|0.480|
> |NGBoost|0.250|0.323|0.362|
> |XGBoost|0.230|0.332|0.360|
>
> We see similar trends as in Figure 5, but there are a few interesting insights: while NAO had the highest KT for 20, 60, and 100 iterations, GCN had the steepest improvement and BONAS had the smallest improvement.
>
> Thank you for this idea which has led to an even better comparison of the model-based performance predictors. Please let us know if you have additional questions or concerns.

---

### Author Response · Authors · 2021-08-14
**Additions following the reviewers’ comments**

Dear reviewers and AC, we have now addressed all of the suggestions and concerns mentioned by the reviewers. We thank all of the reviewers once again, as these suggestions have substantially improved the quality of our paper. We give a list of the main additions below.

- We added a table to clarify for which of the 31 predictors we have succeeded in reproducing the original results, and for which predictors it is not possible. For each predictor, we marked whether the original paper had (1) at least one search space out of the ones we used, (2) at least one initialization and query time that matches one of our settings, and (3) at least one metric from our set of metrics. If yes to all of (1)-(3), then we checked whether we (4) achieved nearly the same numbers as the original paper. We found that of the 31 predictors, 14 were released before any of the NAS-Bench search spaces had come out, and three more did not give rank correlation experiments, but we were able to reasonably reproduce the results from the remaining 14 predictors. See the full details in our [response to reviewer RXE7](https://openreview.net/forum?id=6RB77-6-_oI&noteId=LptZOJcG7mN).
- __update Aug 18__: we have now compared 15 predictors to their original paper, and all 15 are within 0.04 of the original rank correlation result or higher.
- We added two new performance predictors suggested by reviewer Z9bu, and two zero-cost baselines suggested by reviewer eo6y. One of the new predictors, "LCE using previous builds", outperforms the other hybrid predictor, LcSVR, in several settings. Also, surprisingly the new zero-cost baselines (params and flops) outperformed the majority of the other zero-cost predictors.
- We ran an additional experiment that increased the HPO budget from 15 minutes to 1 hour on NAS-Bench-201 cifar10. The goal of this experiment was to identify predictors which may have not been performing well due to insufficient tuning. We found that a few of the deep learning based predictors improved performance by a modest amount. See the details in our [response to reviewer RXE7](https://openreview.net/forum?id=6RB77-6-_oI&noteId=LptZOJcG7mN).
- While our paper already included the mean and standard deviation across different random seeds, we included a new experiment in which we computed the standard deviation of the predictors’ performance when train and test sets are fixed. This gives useful information on the stochasticity of the predictor itself. See the details in our [response to reviewer Z9bu](https://openreview.net/forum?id=6RB77-6-_oI&noteId=c-0rneHkZ56).
- We added plots for OMNI-SemiNAS in the setting of Figure 3, and we added OMNI-NGB and OMNI-SemiNAS to the setting of Figure 4.

If the reviewers have any more comments or concerns, please let us know. Thank you for your time!

---

### Decision · Program_Chairs · 2021-09-27

**Decision:**

Accept (Poster)

**Comment:**

This is an in-depth investigation of existing techniques. The experiments support the claims/questions in the paper well. The authors engaged well during the discussion phase to clarify all reviewers' concerns. The findings are likely going to be very useful to the NAS community.